# Molecular patterns of resistance to immune checkpoint blockade in melanoma

Martin Lauss[1,2], Bengt Phung[1,2], Troels Holz Borch[3], Katja Harbst [1,2], Kamila Kaminska[1,2], Anna Ebbesson[1,2], Ingrid Hedenfalk [1,2], Joan Yuan[4], Kari Nielsen [2,5], Christian Ingvar[6], Ana Carneiro[1,7], Karolin Isaksson[2,6,8], Kristian Pietras [2,9], Inge Marie Svane [3], Marco Donia [3,10] & Göran Jönsson [1,2,10] ✉

Immune checkpoint blockade (ICB) has improved outcome for patients with metastatic melanoma but not all benefit from treatment. Several immune- and tumor intrinsic features are associated with clinical response at baseline. However, we need to further understand the molecular changes occurring during development of ICB resistance. Here, we collect biopsies from a cohort of 44 patients with melanoma after progression on anti-CTLA4 or anti-PD1 monotherapy. Genetic alterations of antigen presentation and interferon gamma signaling pathways are observed in approximately 25% of ICB resistant cases. Anti-CTLA4 resistant lesions have a sustained immune response, including immune-regulatory features, as suggested by multiplex spatial and T cell receptor (TCR) clonality analyses. One anti-PD1 resistant lesion harbors a distinct immune cell niche, however, anti-PD1 resistant tumors are generally immune poor with non-expanded TCR clones. Such immune poor microenvironments are associated with melanoma cells having a de-differentiated phenotype lacking expression of MHC-I molecules. In addition, anti-PD1 resistant tumors have reduced fractions of PD1$^+$ CD8$^+$ T cells as compared to ICB naïve metastases. Collectively, these data show the complexity of ICB resistance and highlight differences between anti-CTLA4 and anti-PD1 resistance that may underlie differential clinical outcomes of therapy sequence and combination.

Immune checkpoint blockade (ICB) therapy has had a major clinical success in advanced stage melanoma. Objective response rates for anti-CTLA4, anti-PD1 and combination therapy of anti-CTLA4 with anti-PD1 were 19%, 45% and 58%, respectively[1]. Despite the clinical progress a large fraction of patients with melanoma will not benefit from ICB.

The majority of non-responders are primary resistant, and a smaller fraction acquires resistance to ICB during treatment. Primary resistance manifests shortly after treatment and is accompanied by progressive disease (PD), whereas acquired resistance is observed after a period of time with initial complete response (CR) or partial response

[1]Division of Oncology, Department of Clinical Sciences, Faculty of Medicine, Lund University, 22185 Lund, Sweden. [2]Lund University Cancer Center, LUCC, Lund, Sweden. [3]National Center for Cancer Immune Therapy, Department of Oncology, Copenhagen University Hospital, Herlev, Denmark. [4]Division of Molecular Hematology, Department of Laboratory Medicine, Faculty of Medicine, Lund University, 22185 Lund, Sweden. [5]Division of Dermatology, Skåne University Hospital and Department of Clinical Sciences, Faculty of Medicine, Lund University, 22185 Lund, Sweden. [6]Division of Surgery, Department of Clinical Sciences, Faculty of Medicine, Lund University, 22185 Lund, Sweden. [7]Department of Hematology, Oncology and Radiation Physics, Skåne University Hospital Comprehensive Cancer Center, 22185 Lund, Sweden. [8]Department of Surgery, Kristianstad Hospital, 29133 Kristianstad, Sweden. [9]Division of Translational Cancer Research, Department of Laboratory Medicine, Faculty of Medicine, Lund University, 22185 Lund, Sweden. [10]These authors contributed equally: Marco Donia, Göran Jönsson. ✉e-mail: goran_b.jonsson@med.lu.se

(PR)[2]. In principle, factors leading to disease progression can be pre-existing at baseline, acquired genetically, or adapted non-genetically, with possible interplay between these resistance mechanisms[3,4].

In baseline pre-treatment samples, a wide range of factors that predict ICB outcome has been reported[5–8]. Tumor-cell intrinsic factors include tumor mutational burden[8–10], mutational subsets[11], clonality[12], aneuploidy[13], immune evasion[14,15], antigen presentation[16,17] and interferon gamma signaling[10,18–20]. Most other predictive factors derive from T cell immunity, such as presence and infiltration of CD8[+] T cells[21], cytotoxicity[9], expression of T cell checkpoints[22], T cell receptor repertoire[23] and T cell sub-populations[24], in particular naïve T cells expressing *TCF7*[25]. Yet, additional cells from the tumor microenvironment can modulate ICB outcome, such as B cells via the formation of tertiary lymphoid structures[26], or fibroblasts via immune cell exclusion[27]. Notably, the constitution of the gut and tumor microbiome affects therapy outcome[28]. In contrast to baseline samples, few ICB resistant samples have been studied so far. Here, loss-of-function (LoF) mutations in *B2M*, *JAK1* and *JAK2*[29], as well as genomic loss of *B2M*[16], alone or co-existing were reported in samples with acquired resistance. In addition, there is evidence of neoantigen loss[30] and T cell re-exhaustion[31] in progressing tumor lesions. With regard to acquired resistance, it is informative that CRISPR screens of tumor cells evading either PD1 blockade or T cell co-culture converge on inactivation of two pathways: interferon-gamma signaling and MHC-I antigen presentation[32–34].

Despite these efforts, a full picture of the molecular mechanisms explaining ICB resistance is lacking, due to a paucity of tumor samples available at or after ICB progression. In addition, it is unclear whether CTLA4- and PD1 blockade resistant samples are substantially different. In this work, we undertook a comprehensive molecular exploration of tumor intrinsic and immune microenvironmental features to further unravel resistance to PD1 and CTLA4 blockade.

## Results

### Genetic analysis of melanoma metastases resistant to ICB

To dissect molecular alterations associated with tumors resistant to different ICB regimens, we have collected tissue samples at progression on ICB treatment at the national Center for Cancer Immune Therapy (CCIT-DK) in Copenhagen, Denmark. Collectively, 23 metastases were from patients resistant to anti-CTLA4 monotherapy (CTLA4res) and 21 metastases from patients resistant to anti-PD1 monotherapy (PD1res). Importantly, all biopsies from metastases were taken after progression on either CTLA4 or PD1 blockade. Moreover, 17 out of the 21 PD1res patients had received and progressed or relapsed on prior anti-CTLA4 treatment. All CTLA4res patients were naïve to PD1 blockade and instead, the majority of CTLA4res patients had received prior IL-2 treatment. Seven patients had also relapsed on BRAF inhibitor (BRAFi). In addition, biopsies from 11 PD1res patients were taken at day 7 during treatment with BRAFi according to a study protocol (PD1res*). Moreover, biopsy from one CTLA4res patient was taken at day 7 during BRAFi treatment[35]. Most patients displayed primary resistance to ICB, except six cases that clinically had acquired resistance (Table 1). Site of primary melanoma was cutaneous skin or unknown primary, except three cases from patients with primary mucosal melanoma that were resistant to anti-CTLA4 (Table 1). Metastatic lesions from patients with mucosal melanoma were excluded from downstream statistical analyses due to their distinct biological characteristics[36]. Hence, we believe that that resistance mechanisms in such melanomas may be different from cutaneous melanomas. Using whole-exome sequencing data, tumor mutational burden, defined here by the total number of non-silent somatic mutations, was not different between CTLA4res and PD1res tumors. Moreover, we compared our data to distant metastases from the cancer genome atlas (TCGA) cohort ($n = 68$), where prior systemic therapy did not include ICB (Fig. 1A)[37], and to two public datasets at ICB baseline

**Table 1 | Clinical characteristics of immune checkpoint blockade (ICB) resistant melanoma**

| *n* (%) | | CTLA4res 20 (45) | PD1res 21 (48) | Mucosal[#] 3 (7) | *p*-value[§] |
|---|---|---|---|---|---|
| Age (median, range) | | 53 (25–68) | 54 (42–73) | 49 (46–65) | 0.13 |
| Gender, *n* (%) | Female | 11 (55) | 10 (48) | 2 (67) | 0.76 |
| | Male | 9 (45) | 11 (52) | 1 (33) | |
| Primary Type | skin | 17 (85) | 17 (81) | 0 (0) | 1 |
| | unknown | 3 (15) | 4 (19) | 0 (0) | |
| | mucosal | 0 (0) | 0 (0) | 3 (100) | |
| Stage | IIIb | 0 (0) | 1 (5) | 1 (33) | 1 |
| | M1a | 2 (10) | 1 (5) | 0 (0) | |
| | M1b | 1 (5) | 2 (10) | 0 (0) | |
| | M1c | 17 (85) | 17 (81) | 2 (67) | |
| Previous BRAFi | yes | 4 (20) | 3 (14) | 0 (0) | 0.70 |
| | no | 16 (80) | 18 (86) | 3 (100) | |
| Previous CTLA4i | yes | 20 (100) | 17 (81) | 3 (100) | – |
| | no | 0 (0) | 4 (19) | 0 (0) | |
| Previous IL2 | yes | 18 (90) | 2 (10) | 3 (100) | 2 x 10⁻⁷ |
| | no | 2 (10) | 19 (20) | 0 (0) | |
| Biopsy taken during BRAFi treatment* | yes | 1 (5) | 11 (52) | 0 (0) | 0.001 |
| | no | 19 (95) | 10 (48) | 3 (100) | |
| Resistance Type | primary | 15 (75) | 15 (71) | 3 (100) | 1[§§] |
| | primary (ipi acq.) | 0 (0) | 3 (14) | 0 (0) | |
| | acquired | 3 (15) | 3 (14) | 0 (0) | |
| | NA | 2 (10) | 0 (0) | 0 (0) | |
| WES | yes | 17 (85) | 17 (81) | 3 (100) | - |
| | no | 3 (15) | 4 (19) | 0 (0) | |
| RNAseq | yes | 18 (90) | 20 (95) | 3 (100) | - |
| | no | 2 (10) | 1 (5) | 0 (0) | |

Patients are divided into anti-CTLA4 resistant (CTLA4res), anti-PD1 resistant (PD1res) and mucosal melanoma.
*biopsy taken at day 7 with BRAFi
[#]all mucosal melanomas only received anti-CTLA4 and displayed primary resistance
[§]CTLA4res vs PD1res, using Fisher's exact test, except for Age using Wilcoxon test
[§§]Excluding NAs and primary (ipi acq.)

(Supplementary Fig. 1A)[5,38], and did not observe any differences in mutational burden. Together, mutational frequencies were similar between metastases progressing or relapsing on either anti-CTLA4 or anti-PD1 and ICB naïve melanomas.

As expected, samples from ICB resistant patients contained *BRAF* V600 ($n = 24$, 71%, not including mucosal samples) and *NRAS* Q61 ($n = 7$, 21%) mutations in a mutual exclusive way (Fig. 1B). Moreover, *CDKN2A* had the highest frequency of inactivation with nine deep deletions and four loss-of-function (LoF) mutations in ICB resistant cases ($n = 13$, 34%) however, this frequency was not different as compared to the treatment-naïve distant metastases from TCGA (Fig. 1B, Supplementary Fig. 1B). Other known melanoma driver genes including *TP53* ($n = 4$, 12%) and *PTEN* ($n = 3$, 9%) also harbored inactivating events in ICB resistant cases. Driver mutations were predominantly clonal (Supplementary Fig. 2A). In a genome-wide analysis, we did not observe novel genes with high frequencies of alterations ($n >= 4$ hotspot mutation, LoF mutation, deletion or amplification) in ICB resistant cases (Supplementary Fig. 2B–D). Interestingly, *APC* and *PLCB4* had three nonsense mutations each.

In summary, the frequencies of genetic alterations in melanoma driver genes were not different in anti-CTLA4 and anti-PD1 resistant and ICB naïve metastatic melanomas.

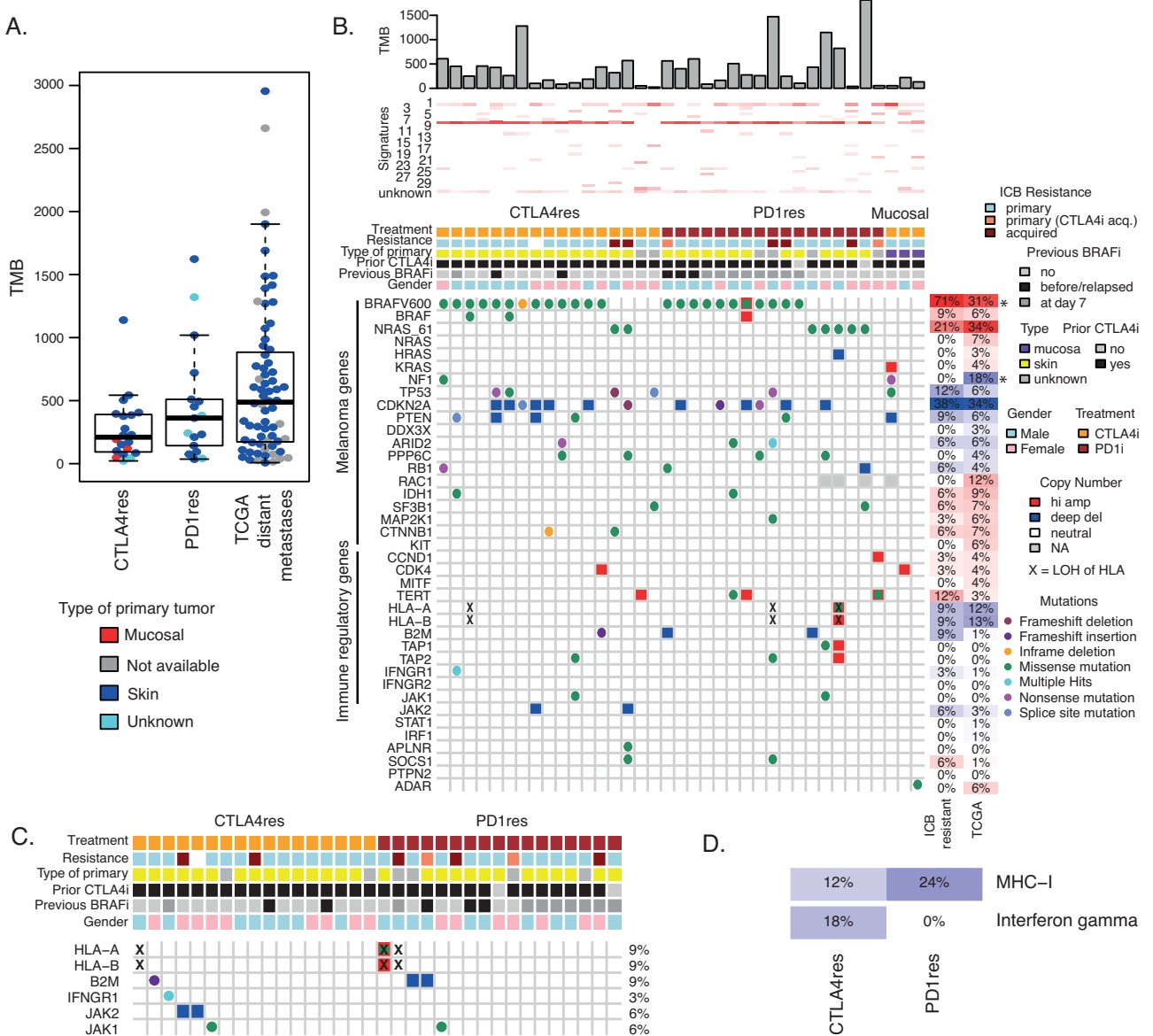

**Fig. 1 | Mutational landscape of immune checkpoint blockade (ICB) resistant melanoma. A** Tumor mutational burden (TMB) calculated as total number of somatic mutations in n = 20 anti-CTLA4 resistant (CTLA4res) or n = 17 anti-PD1 resistant tumors (PD1res) as compared to n = 68 ICB naïve distant metastases from the Cancer Genome Atlas (TCGA) n = 68. Boxplot is displayed with the center-line as median, the box limits as lower and upper quartiles, and with whiskers covering the most extreme values within 1.5 x Interquartile-Range. **B** Genetic aberrations of selected genes in CTLA4res (n = 17) and PD1res (17) resistant melanomas, combining mutation-, copy number- and HLA Loss-of-Heterozygosity (LOH) levels. The majority of PD1res melanomas had previously relapsed on CTLA4 blockade. Tumor mutational burden and mutational signatures are indicated on top. Frequencies of activating events for potential oncogenes and loss-of-function events for potential

tumor suppressor genes are depicted on the right for ICB resistant tumors excluding mucosal samples, and for the ICB naïve TCGA control cohort, respectively, and significant differences are indicated by * (BRAF P = 3 × 10⁻⁴ and NF1 P = 0.008, Fisher test). All test were two-sides. Three mucosal melanomas are displayed in heatmap but excluded from statistical analyses. **C** Genetic alterations of genes in the interferon gamma and MHC-I pathways between CTLA4res (n = 17) and PD1res (n = 17). The frequency of combined events is noted for each gene. Annotation and event legends as in **B. D** Frequency plot of immune regulatory pathways in CTLA4res (n = 17) and PD1res (n = 17) melanomas, considering only loss-of-function events. Amp Amplification. Del Deletion. Source data and exact p-values are provided as a Source Data file.

## Genetic alterations in immune regulatory pathways

Next, we specifically analyzed genetic alterations occurring in immune regulatory pathways. Previously, loss of B2M was reported to be associated with resistance to ICB in melanoma and is essential for HLA class I assembly and presentation on the cell surface[16]. In this study, one CTLA4res case had a frameshift deletion and two PD1res lesions had deep deletions of *B2M* that consequently also had loss of the B2M protein in the tumor cells (Supplementary Fig. 3). Moreover, *HLA-A, HLA-B,* or *HLA-C* lacked LoF mutations, however, the HLA locus had LOH in three ICB resistant cases (9%), one in CTLA4res and two in

PD1res patients (Fig. 1C). Together antigen presentation was impaired in 18% (n = 6) of ICB resistant tumors of which 12% of the CTLA4res and 24% of the PD1res (Fig. 1D). In comparison, the treatment-naïve TCGA data had LOH at the *HLA* locus in 13% of cases, and *B2M* and *TAP2* had one LoF mutation each, resulting in a similar frequency of *MHC-I* inactivation (Supplementary Fig. 1B). In addition, in the interferon-gamma pathway that has been implicated in ICB resistance[19], we found two deep deletions of the *JAK2* gene and a frameshift mutation in the *IFNGR1* gene, all cases being CTLA4res. We also compared tumor biopsies from patients that clinically demonstrated intrinsic or

acquired resistance and did not find any differences (Supplementary Fig. 1C). Further, we compared PD1res patients with prior relapse to anti-CTLA4, to anti-PD1 naïve samples with prior relapse to anti-CTLA4 from two public datasets[5,38] to isolate the effect of PD1 blockade. However, we found similar mutational landscapes across the datasets (Supplementary Fig. 1D).

Together, genetic alterations in genes belonging to antigen presentation- or interferon-gamma pathways occurred in different samples, however in total only accounting for 26% of ICB resistant cases suggesting that other still unknown immune evasive mechanisms exist.

## Immune transcriptional programs are different in anti-CTLA4 and anti-PD1 resistant melanomas

As the genetic landscape in the cohort only explained a minority of ICB resistance we performed transcriptomic profiling using RNA sequencing. In total, 17 (non-mucosal) melanoma metastases from anti-CTLA4 resistant (CTLA4res) and 21 from anti-PD1 resistant (PD1res) cases were analyzed. Anti-PD1 resistant metastases taken during BRAFi treatment were treated as a separate group (PD1res*) as previous studies have demonstrated an influx of immune cells in tumors during BRAFi treatment[39]. The single CTLA4res patient where biopsy was taken at day 7 during BRAFi treatment was excluded from downstream statistical analyses. Indeed, in this cohort, immune cell signatures[40,41] were increased in BRAFi treated tumors and this was specifically pronounced when comparing within the anti-PD1 resistant tumors alone (Fig. 2A). Moreover, the cell cycle module was upregulated in PD1res tumors ($P = 0.006$) whereas the immune module was upregulated in CTLA4res melanomas ($P = 0.02$). In addition, a variety of immune cell type signatures[15,25,41], such as T-cells, NK-cells, monocytes and dendritic cells, had higher scores in CTLA4res melanomas (Fig. 2A, Supplementary Fig. 4A). Exhaustion signatures were also at higher levels in CTLA4res melanomas, probably due to a higher infiltration of immune cells. Further, we also observed single gene expression of MHC-I, MHC-II, interferon gamma signaling, Tumor/T-cell interaction, inflammation and cytotoxicity genes, all generally at higher levels in CTLA4res melanomas, albeit not always crossing significance thresholds (Supplementary Fig. 4B). In contrast, immune exclusion genes, e.g., *MYC*, demonstrated an upregulation in PD1res melanomas. Gene set enrichment analysis of discriminating genes confirmed that cell cycle processes, such as DNA replication, were elevated in PD1res tumors (FDR < 0.001) and immune processes, specifically from the adaptive immune system, were elevated in the CTLA4res cases (FDR < 0.001) (Fig. 2B). However, CD3 immunofluorescence staining could not confirm a statistically increased frequency of CD3$^+$ T cells in the CTLA4res tumors ($P = 0.12$) (Fig. 2C). Finally, we used T cell receptor clonotype sequencing and found that CTLA4res tumors had a higher TCR clone abundance as well as a higher frequency of patients with dominant TCR clones compared to PD1res melanoma ($P = 0.047$, Fig. 2D), suggesting a higher number of tumor-reactive T cells in the CTLA4res cases. Moreover, TCR clonality results demonstrated that PD1res* had an increased clonotype count, which is also supported by abundance of CD3$^+$ T cells in such tumors. In contrast, PD1res* did not show an increase in clonal expansion as compared to PD1res melanomas (Fig. 2D). Thus, TCR clonality analysis suggests that BRAFi is associated with a higher abundance of T cells in melanomas but not expansion of specific TCR clones.

To understand differences between ICB regimens with respect to tumor intrinsic properties, we selected tumor cell regions using S100B/PMEL antibodies for digital spatial profiling. Most genes in the panel were upregulated in CTLA4res as compared to PD1res melanomas with *PMEL* being one of the most differentially expressed genes ($P = 0.1$, Fig. 2E). This suggests that anti-PD1 resistance correlates with decreased melanocytic antigens which may facilitate immune escape.

In summary, the observations indicate a sustained immune response in some tumors despite progressing on anti-CTLA4; in contrast, a particularly immune-poor microenvironment was observed in anti-PD1 resistant melanoma.

## Distinct tumor cell states exist in melanoma metastases

Several intrinsic tumor cell states have been suggested for melanoma, which generally align on a gradient of MITF-low to MITF-high expression levels[42–44]. These cell states can switch to adapt to external cues. The association of such melanoma cell states to the immune microenvironment and immunotherapy resistance has not been thoroughly investigated. We therefore performed Visium sequencing on six melanoma metastases (three CTLA4res, one PD1res, one anti-CTLA4 resistant mucosal and one ICB naïve), and defined melanoma cell states as characterized by differential expression of *MITF*, *MKI67*, *NGFR*, *AXL* and *TAP1*. Consensus clustering of 2,766 tumor cell-enriched spots resulted in five groups where four were characterized mainly by differential expression of *MITF* and *TAP1* (Fig. 3A). The fifth group had decreased levels of *MITF* and increased levels of *NGFR* gene expression. By morphologically mapping such melanoma states on H&E stainings we found an extensive heterogeneity of melanoma cell states in all ICB resistant tumors (Fig. 3B), whereas the ICB naïve metastasis harbored predominantly *MITF*$^{high}$/*TAP1*$^{high}$ melanoma cells suggesting an immunogenic state, which was supported by numerous spots with increased gene expression of immune cell markers across that tumor section (Supplementary Fig. 5). To expand on these findings, we used multiplex immunofluorescence (mIF) staining of 10 CTLA4res and 15 PD1res metastatic lesions, of which seven were PD1res*, and added staining on metastases from 53 patients with stage IV ICB naïve melanoma. First, we could confirm the results from the digital spatial profiling analysis showing a decreased expression of MITF in PD1res melanomas, however not reaching statistical significance when compared to CTLA4res or ICB naïve melanomas (Fig. 3C). Notably, one of the PD1res MITF$^{high}$ melanomas harbored a *B2M* genetic alteration. The percentage of NGFR$^+$ melanoma cells was not associated with ICB resistance ($P = 0.96$) as very few melanoma tumors harbored notable fractions of NGFR$^+$ melanoma cells. We then mimicked the melanoma cell states identified by the Visium sequencing using mIF antibodies for SOX10, MITF, B2M and NGFR. As expected, we found a correlation between presence of B2M$^+$/MITF$^{high}$ melanoma cell populations and CD8$^+$ (Spearman cor. 0.75, $P < 0.0001$) and CD3$^+$ T cell abundance (Spearman cor. 0.66, $P < 0.0001$). PD1res tumors were depleted of MITF$^{high}$B2M$^+$ melanoma cell populations and enriched in MITF$^{low}$B2M$^-$ populations (Fig. 3D).

In conclusion, melanoma metastases harbor multiple tumor cell populations that are associated with T cell infiltration.

## B cells and tertiary lymphoid structures are rare in anti-PD1 resistant metastatic melanomas

Tertiary lymphoid structures (TLS) are ectopic immune cell niches with resemblance to secondary lymphoid organs[45]. We and others recently observed TLSs in ICB naïve melanoma metastases and found that such structures correlated to patient survival and ICB clinical response[26,46]. Here, in line with CD3$^+$ T cell abundance, we observed a higher frequency of CD20$^+$ B cells in CTLA4res lesions as compared to PD1res lesions that contained very few CD20$^+$ B cells ($P = 0.06$). When compared to melanoma metastases from ICB naïve patients, no significant difference was found, most likely due to the vast heterogeneity of CD20$^+$ B cell presence observed in ICB naïve cases (Fig. 4A). Two CTLA4res metastases harbored multiple immature TLSs, while six ICB naïve metastases had at least one TLS which appeared as more mature than TLSs found in CTLA4res metastases (Fig. 4B, C). None of the PD1res metastatic melanomas had TLSs. However, one PD1res melanoma harbored an increased frequency of CD20$^+$ B cells, which were not organized in TLS and instead were scattered and localized at the tumor margin (Fig. 4D). Intriguingly, this PD1res melanoma had a massive infiltration of CD3$^+$ T cells and had predominantly B2M

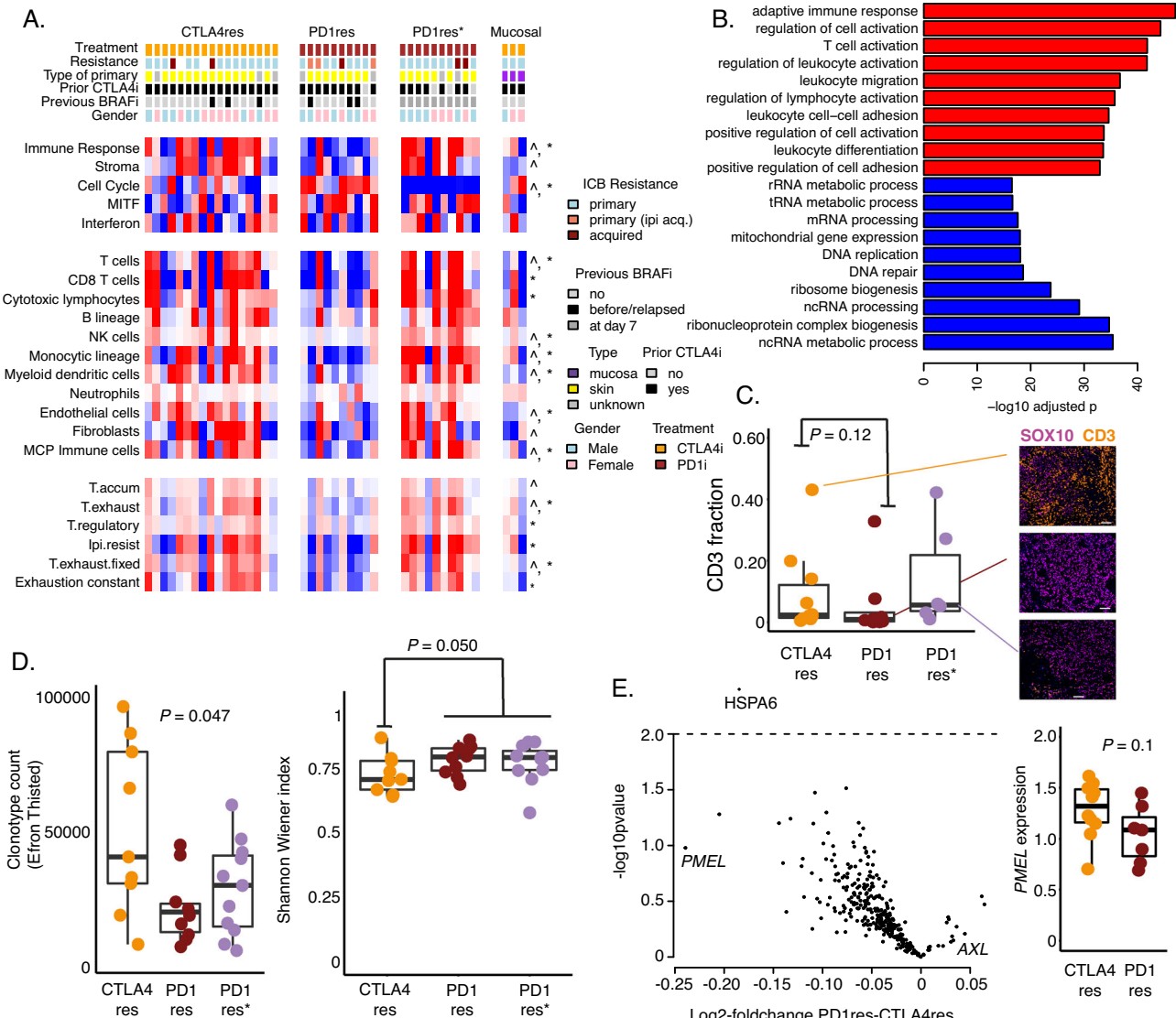

**Fig. 2 | The immune transcriptomic landscape of immune checkpoint blockade (ICB) resistant melanoma. A** Heatmap of melanoma module- and immune pathway transcriptomic scores[40,41,86,87]. Patient samples are divided according to anti-CTLA4 resistant (CTLA4res, $n = 17$), anti-PD1 resistant (PD1res, $n = 10$) and anti-PD1 resistant biopsies taken at day 7 during BRAFi treatment (PD1res*, $n = 10$). *$P < 0.05$ between anti-PD1 without and under BRAFi treatment. ^$P < 0.05$ between anti-PD1 without BRAFi and anti-CTLA4 resistant lesions. T.accum – accumulated T cell score, T.exhaust – exhausted T cell score, T.regulatory – regulatory T cell score, Ipi.resist – signature score associated with anti-CTLA4 resistance, T.exhaust.fixed – exhausted/fixed T cell score. *P*-values from t-test. All tests were two-sided. Exact p-values are provided in Source data. Three mucosal melanomas are displayed in heatmap but excluded from statistical analyses. **B** Top ten pathways from gene set enrichment analysis of genes differentiating anti-CTLA4 from anti-PD1 resistant melanomas. This analysis was conducted on genes derived from the DESeq2 analysis. *P*-values from gene set enrichment analysis and adjusted for multiple testing. Red = enriched in anti-CTLA4 group, blue = enriched in anti-PD1 group. **C** Fraction

of CD3+ cells as determined by multiplex immunofluorescence in $n = 10$ CTLA4res, $n = 8$ PD1res and $n = 6$ PD1res* tumors. Representative images are shown. Scalebar is indicated by white line and corresponds to 100 μm. *P*-value from Wilcoxon test. Test was two-sided. **D** Boxplots of T cell receptor (TCR) clonality data in $n = 9$ CTLA4res, $n = 9$ PD1res and $n = 11$ PD1res* tumors. Left boxplot shows the clonotype count (Efron Thisted). *P*-value from Kruskal-Wallis text. Right boxplot shows the evenness according to normalized Shannon Wiener index. *P*-value from Wilcoxon test. **E** Spatial gene expression data of tumor cell regions using Nanostring GeoMx, normalized for SOX10 expression. Volcano plot showing differentially expressed genes between $n = 10$ anti-CTLA4 and $n = 7$ anti-PD1 resistant tumors. Boxplot of *PMEL* expression between the two groups. *P*-values from t-test. Test was two-sided. Boxplots are displayed with the center-line as median, the box limits as lower and upper quartiles, and with whiskers covering the most extreme values within 1.5 x Interquartile-Range. Source data and exact p values are provided as a Source Data file.

positive melanoma cells in contrast to most other PD1res melanomas. To further understand lymphocyte phenotypes in ICB resistant and naïve metastatic melanomas we performed single cell RNA sequencing of four tumors with TLS or B cells (1 CTLA4res, 1 PD1res and 2 ICB naïve). The 26,053 sequenced cells stemmed from a wide range of cell types including B cells (Fig. 5A). Using a set of B cell and TLS specific genes (Supplementary Table 1), clustering revealed eight distinct B cell clusters: four plasma cell, two naïve B cell, one germinal center-like B

cell and one memory B cell-like group (Fig. 5B). The ICB naïve 1 metastasis contained predominantly naïve B cells and germinal center-like B cells, suggesting presence of highly mature TLSs, whereas the ICB naïve 2 metastasis contained mainly plasma cells together with additional B cell phenotypes (Fig. 5C). The CTLA4res metastatic melanoma consisted of naïve and memory-like B cells and only smaller fractions of plasma cells. Finally, the PD1res metastatic melanoma, with B cells and massive T cell infiltration, had predominantly (>80%)

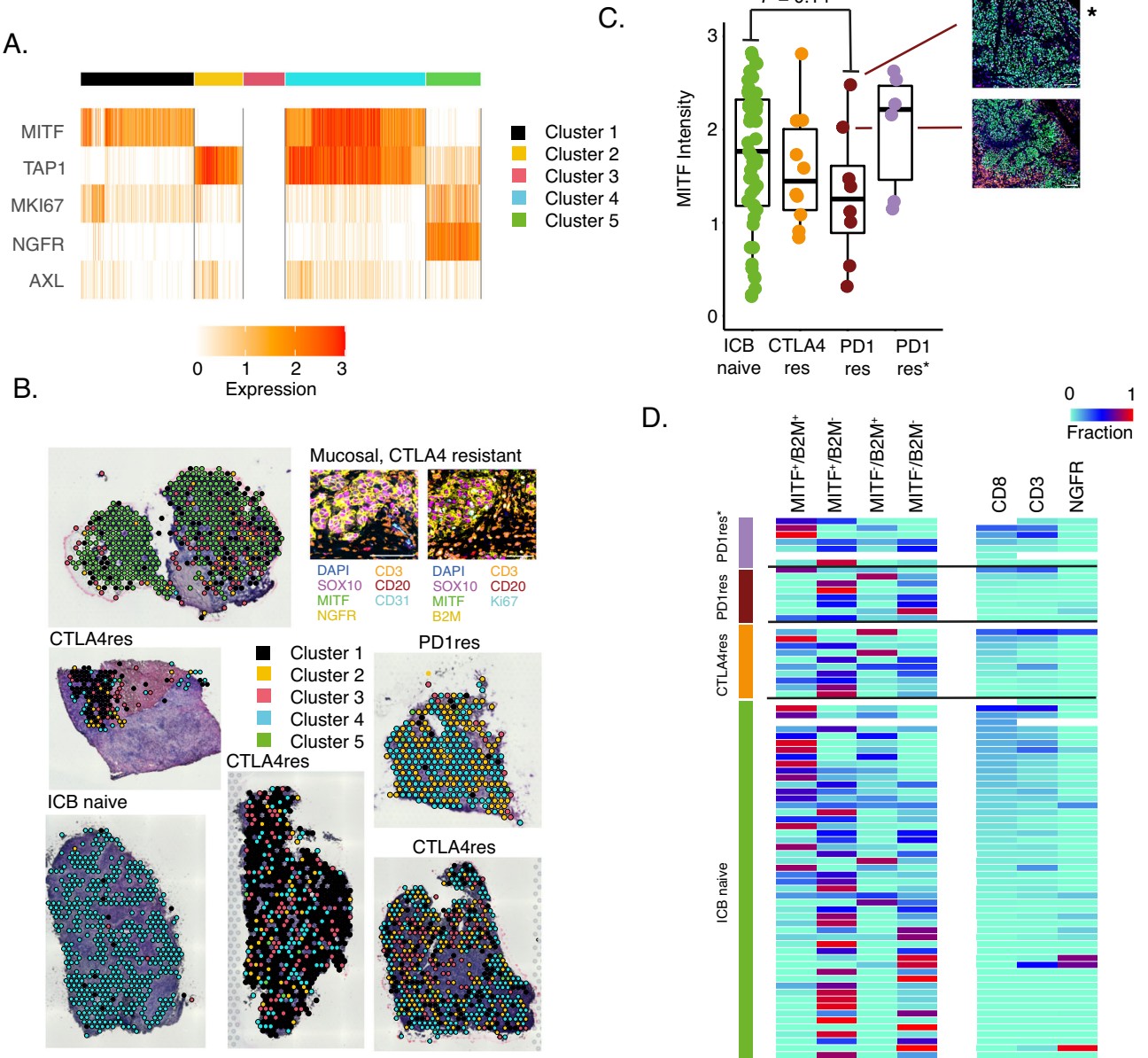

**Fig. 3 | Tumor cells states in immune checkpoint blockade (ICB) resistant melanoma. A** Heatmap displaying expression of five melanoma state specific genes (*MITF, TAP1, MKI67, NGFR, AXL*) across 2,766 tumor cell enriched spots from the Visium data in six melanoma tumors. Spots were divided in five distinct clusters based on consensus clustering and are grouped by this cluster assignment. **B** Mapping back melanoma cell clusters as defined in **A**, from six melanoma metastases, to the respective histological images. Indicated is also a validation using multiplex immunofluorescence. **C** MITF multiplex immunofluorescence intensity of *n* = 51 ICB naïve, *n* = 10 anti-CTLA4 resistant (CTLA4res), *n* = 8 anti-PD1 resistant (PD1res) and *n* = 6 anti-PD1 resistant under BRAFi treatment (PD1res*).

MITF intensity was measured in SOX10 positive melanoma cells. *P*-value from Wilcoxon test. Test was two-sided. * denotes a melanoma with an MITF-high phenotype and B2M deep deletion. Scalebars are indicated by white line and correspond to 100 μm. Boxplot is displayed with the center-line as median, the box limits as lower and upper quartiles, and with whiskers covering the most extreme values within 1.5 x Interquartile-Range. **D** Frequency plot of different melanoma cell states using multiplex immunofluorescence of MITF/B2M fractions and NGFR fractions. NGFR fractions are within SOX10 positive melanoma cells. Samples are sorted by CD8 fraction and grouped according to treatment. (PD1res *n* = 8, PD1res* *n* = 7, CTLA4res *n* = 10, ICB naïve *n* = 53). Source data are provided as a Source Data file.

memory-like B cells. Memory B cells from the PD1res melanoma were mainly IgA⁺ cells, in contrast to the samples with conventional TLS, which had large fractions of IgG⁺ memory B cells (Fig. 5D). Indeed, *IgA* expressing B cells have been reported to have immunosuppressive consequences by inducing distinct T cell phenotypes[47]. With this in mind, we observed 11 T cell clusters in the single cell RNA sequencing data, including naïve T cells, T follicular helper cells, T regulatory cells and effector/exhausted T cells (Fig. 5E). Strikingly, T cells from the PD1res melanoma consisted almost exclusively of effector/exhausted CD8⁺ T cells (Supplementary Fig. 6 A). Some of the CD8⁺ T cells

expressed *PD1*, they however lacked *TCF7* expression (Fig. 5F), suggesting a lack of a replenishing T cell reservoir.

Altogether, scRNAseq data reveal distinct B cell phenotypes in ICB resistant metastatic melanomas that may be linked to T cell phenotype.

### Distinct T cell phenotypes are enriched in ICB resistant metastatic melanomas
Consequently, we went on to investigate T cell presence and different phenotypes in ICB resistant compared to ICB naïve cases using mIF. A

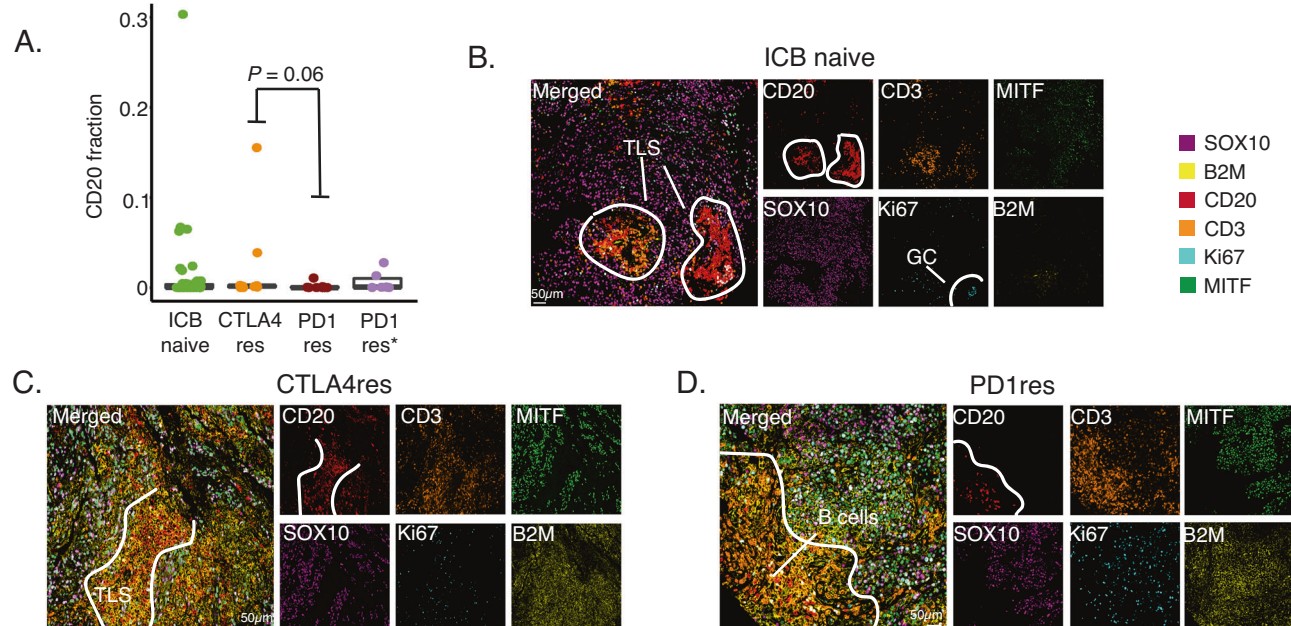

**Fig. 4 | B cells and tertiary lymphoid structures in immune checkpoint blockade (ICB) resistant melanoma. A** Boxplot of fractions of CD20 + B cells by treatment using multiplex immunofluorescence. *P*-value from Wilcoxon test. Test was two-sided. Green - ICB naïve (*n* = 52), orange - anti-CTLA4 resistant (CTLA4res) (*n* = 10), red - anti-PD1 resistant (PD1res) (*n* = 8), purple - anti-PD1 resistant under BRAFi (PD1res*) (*n* = 6). Boxplot is displayed with the center-line as median, the box limits as lower and upper quartiles, and with whiskers covering the most extreme values within 1.5 x Interquartile-Range. **B**–**D** Multiplex immunofluorescence images of TLSs and B cells in ICB naïve (**B**), CTLA4res (**C**) and PD1res (**D**) melanoma. Representative images are shown. Source data are provided as a Source Data file.

wide range of CD3⁺ and CD8⁺ T cell abundance was observed in ICB naïve metastatic melanomas (Fig. 6A), and no significant difference was observed between ICB resistant and naïve cases. However, the majority of PD1res melanomas had very few infiltrating T cells, whereas abundance of CD3⁺ and CD8⁺ T cells in CTLA4res tumors was indistinguishable from ICB naïve melanomas. Immunosuppressive T regulatory cells are specifically characterized by expression of the transcription factor FOXP3. Intriguingly, we found an increase of FOXP3⁺ T cells in CTLA4res tumors as compared to PD1res and ICB naïve melanomas (*P* = 0.047, Fig. 6B). Such FOXP3⁺ T cells were closely co-localized with CD8⁺ T cells (Fig. 6C).

Increased frequencies of TCF7⁺ CD8⁺ T naïve/stem cells have recently been associated with an improved clinical response to ICB[25]. Moreover, tumor-associated T cells lacking PD1 and TCF7 expression are suggested to be bystander T cells, specific to tumor-unrelated targets[48]. Therefore, we classified CD8⁺ T cells using a combination of TCF7 and PD1 expression, using mIF (Fig. 6D, E). As expected, PD1⁺ cells were more proliferating (Ki67⁺) than double positive and TCF7⁺ CD8⁺ T cells[49]. Moreover, the double negative CD8⁺ T cells also had elevated proliferation rates (Supplementary Fig. 6B). We found PD1res melanomas to have a significantly lower frequency of PD1⁺ CD8⁺ T cells (*P* = 0.03) and consequently an increased frequency of double negative (PD1⁻/TCF7⁻) CD8⁺ T cells (Fig. 6F). No difference of TCF7⁺ or double positive CD8⁺ T cells was observed between CTLA4res and naïve melanomas.

In summary, the majority of CD8⁺ T cells infiltrating PD1res samples do not express PD1 and presumably are not tumor reactive. In contrast, CD8⁺ T cell phenotypes in CTLA4res melanoma were indistinguishable from ICB naïve melanoma; however, instead, an increase of FOXP3⁺ T cells was observed. These results indicate that ICB immune microenvironmental resistance mechanisms are different in anti-PD1 and anti-CTLA4 resistant tumor lesions.

## Discussion

A complete picture of genetic and molecular effector mechanisms explaining ICB resistance has so far not been identified. In this study, we combined analyses of the tumor immune microenvironment and tumor intrinsic features on human melanoma specimens taken at progression from patients receiving PD1 or CTLA4 blockade monotherapy. Notably, the majority of anti-PD1 resistant patients herein had before relapsed on CTLA4 blockade. Previous reports have converged on genetic alterations in two major pathways explaining ICB resistance: the interferon-gamma and antigen presentation pathways[16,29]. Specifically, a landmark study reported *JAK1*, *JAK2* and *B2M* LoF mutations in three of four investigated ICB resistant samples, respectively[29]. Another study highlighted 5 of 12 patients progressing on ICB to harbor *B2M* LoF mutations or LOH[16]. In a recent study, 22 cell lines from 18 patients that had progressed on anti-PD1 or anti-PD1/anti-CTLA4 therapy, contained one *JAK2* and two *B2M* inactivating events and two *HLA* LOH events, next to other potential ICB resistance mechanisms[50]. In the present study, we found only 18% harboring genetic alterations in the major genes within the antigen presentation pathway, and 9% had genetic alterations in genes belonging to the interferon-gamma pathway. Importantly, genetic alteration in either pathway can be sufficient to develop resistance to ICB based on CTLA4 or PD1 targeting[51], and we could not definitely determine differences between patients relapsing on CTLA4 or PD1 blockade. However, immune response transcriptional signatures demonstrated increased expression of such genes in anti-CTLA4 resistant samples. Indeed, CTLA4 blockade has demonstrated an increased influx of T cells in post-treatment samples from patients with melanoma[52]. In this study, tumor infiltrating T cells in anti-CTLA4 resistant samples had a significantly increased number of expanded TCR clones as compared to anti-PD1 resistant samples suggesting that such T cells have an increased tumor-reactivity. Intriguingly, we found an increased FOXP3⁺ T cell abundance in anti-CTLA4 resistant tumor lesions and there are reports describing that the immunosuppressive properties of FOXP3⁺ T cells are dependent on TCR signaling[53] suggesting that the increased TCR clonality reflects an increased immunosuppressive environment mediated by FOXP3⁺ T regulatory cells. Interestingly, patient samples taken during BRAFi treatment also had a high fraction of CD3⁺ T cells but no indication of expansion of distinct TCR clones. Several studies have described that BRAFi induces recruitment of immune cells[39] and

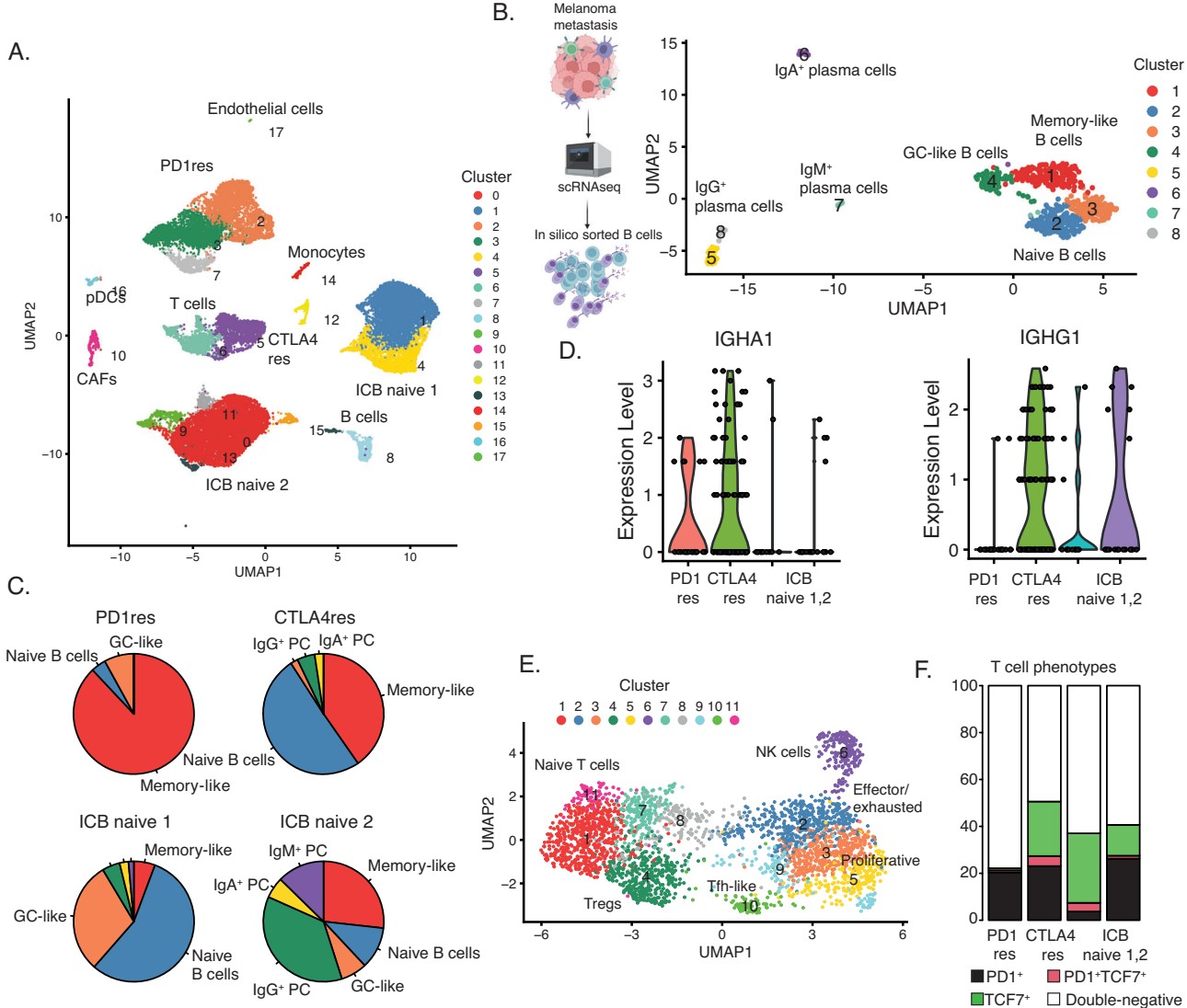

**Fig. 5 | Functional B and T cell phenotype composition using single cell RNA sequencing. A** Uniform Manifold Approximation and Projection (UMAP) plot of 26,053 single cells from one anti-PD1 resistant (PD1res), one anti-CTLA4 resistant (CTLA4res) and two immune checkpoint blockade (ICB) naïve melanomas, with increased B cells. Cell type assignments derived from manual annotation are indicated in the plot. **B** UMAP of 559 B cells visualizing eight distinct clusters. B cell subsets are indicated in the plot after manual annotation. The scheme of the experimental procedure was created with BioRender.com. **C** Pie charts describing the fraction of each B cell subset in the four melanomas. **D** Violin plots of the expression of IGHA1 and IGHG1 in memory-like B cells in the four melanomas. **E** UMAP of 2921 T cells visualizing 11 clusters, in the four melanomas. T cell subsets are indicated in the plot after manual annotation. **F** CD8 T cell phenotype fractions based on non-zero expression of *PD1* and *TCF7* in *CD8A* or *CD8B* expressing T cells of all four melanomas. Source data are provided as a Source Data file.

our data further add evidence to this. The PD1 blockade resistant samples, instead, contained very few T cells. Only a small fraction of the T cell infiltrate is considered to be tumor-reactive, and expression of PD1 is a relevant marker to distinguish tumor-reactive from bystander T cells[48]. In addition, T cells expressing TCF7 were found to be more effectively reinvigorated by ICB and have been associated with improved response to ICB in human melanoma[25]. Here, we found that an inferior number of CD8+/PD1+ T cells are present in anti-PD1 resistant samples. Instead, an increased number of bystander (PD1−/TCF7−) CD8+ T cells were found in anti-PD1 resistant melanomas. Overall, this suggests that the tumor specific immune response is considerably hampered in PD1 resistant cases, which may either have developed during resistance or has pre-existed and was selected for due to the high response rate of PD1 blockade.

One of the strongest predictive biomarkers to ICB across cancer diagnoses is the formation of TLS[26,46]. In this study, we found immature TLSs in anti-CTLA4 resistant tumors, however no TLS was identified in anti-PD1 resistant melanoma. One anti-PD1 resistant melanoma had a massive infiltration of effector/exhausted T cells that was accompanied by spatially scattered IgA+ memory B cells. Indeed, IgA+ B cells have been described to also confer regulatory functions[47], and this tumor may sustain a suppressive immune cell niche. These results demonstrate that we need to know more about the different functional subsets and contexts of tumor-associated lymphocytes.

T cells are activated by exposure to specific antigens. Melanoma tumors can potentially present many different neoantigens as melanoma harbors extensive tumor mutational burden. Further, antigen presentation in tumor cells can be expanded to highly expressed melanocyte differentiation self-antigens, which is regulated by immune tolerance[54]. This renders melanoma to be a potentially highly immunogenic cancer type, however, in some cases immunogenicity can be low due to e.g., reduced neoantigen presentation, or downregulation of melanocyte differentiation antigens[50]. Moreover, several melanoma cell states exist that have different molecular and functional

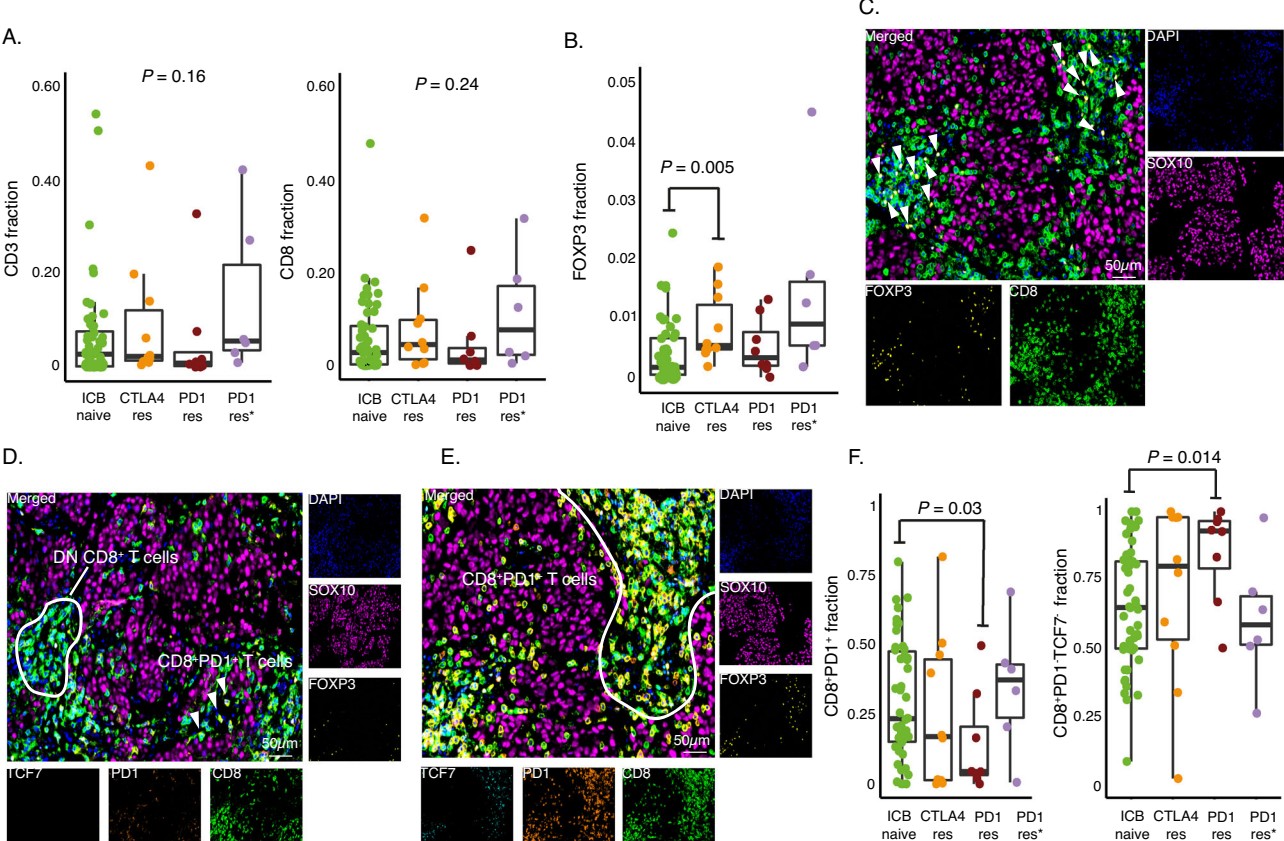

**Fig. 6 | T cell phenotypes in immune checkpoint blockade (ICB) resistant melanomas. A** CD3+ and CD8+ T cell fraction using multiplex immunofluorescence shown as boxplots. *P*-values from Kruskal-Wallis test. **B** FOXP3+ T cell fraction using multiplex immunofluorescence shown as boxplot. P-value from Wilcoxon test. Test was two-sided. **C** Multiplex immunofluorescence images from an anti-CTLA4 resistant (CTLA4res) melanoma that has infiltration of FOXP3+ T cells. FOXP3+ T cells are co-localized with CD8+ T cells and marked by an arrowhead. **D** Multiplex immunofluorescence images from an anti-PD1 resistant melanoma (PD1res) that has a strong infiltration of CD8+ T cells. PD1−/TCF7− double negative CD8+ T cells areas are marked. **E** Multiplex immunofluorescence images from an ICB naive melanoma that has a strong infiltration of CD8+ T cells. PD1+ CD8+ T cells area is

marked. **F** Boxplots of the fraction of PD1+TCF7− of CD8+ T cells and the fraction of double negative T cells (PD1−/TCF7−) using multiplex immunofluorescence. *P*-values from Wilcoxon test. All tests were two-sided. Green - ICB naïve (**A**, **B**, *n* = 52; **F**, *n* = 50) orange - anti-CTLA4 resistant (CTLA4res) (**A**, **B**, **F**, *n* = 10), red - anti-PD1 resistant (PD1res) (**A**, **B**, **F**, *n* = 8), purple - anti-PD1 resistant under BRAFi (PD1res*) (**A**, **B**, **F**, *n* = 6). Boxplots are displayed with the center-line as median, the box limits as lower and upper quartiles, and with whiskers covering the most extreme values within 1.5 x Interquartile-Range. Representative areas from two tumor cores (1 mm in diameter) per melanoma metastasis were selected in the display items in (**C**–**E**). Source data are provided as a Source Data file.

properties[42]. We used spatial transcript sequencing and found five different melanoma cell states. Interestingly, we found a vast heterogeneity of the spatial distribution of the different melanoma states that is similar to previous findings[55]. Our work describes that de-differentiated melanoma cells lacking B2M expression are frequently observed in anti-PD1 resistant melanomas. This is in line with previous reports and indicates that a de-differentiated melanoma state represents a pan-therapy resistance feature[56]. We further demonstrate that such melanoma state is anti-correlated to CD8+ T cell presence suggesting that they escape the recognition by the immune system.

In conclusion, our work provides a comprehensive view of the molecular and immune cell landscape at relapse on ICB. Our data further highlight that development of anti-CTLA4 and anti-PD1 resistance occur through different molecular mechanisms. This study highlights the molecular complexity in development of ICB resistance.

## Methods
### Patients
Patients with ICB resistant melanoma were treated with ICB regimens as per standard of care in Denmark until progression and were subsequently referred to CCIT, Denmark for enrollment in three different clinical trials on adoptive cell therapy[35,57,58]. Sample collection was done

at the time of enrollment in the trials and informed consent was obtained. All three trials (NCT00937625, NCT02379195 and NCT02354690) are listed in clinicaltrials.gov, and all procedures were conducted in accordance with the Declaration of Helsinki and following approval from the Scientific Ethics Committee of the Capital Region of Denmark. Metastatic melanoma lesions from ICB naïve patients were collected at Skåne University Hospital in Sweden prior to clinical introduction of immune checkpoint blockade under the ethical permit Dnr. 101/2013 and 191/2007. Patients signed an informed consent before sample was collected. Clinical data on ICB resistant cases are summarized in Table 1.

### Whole exome sequencing
Whole exome sequencing data were generated as described previously[59] on a NextSeq500 instrument (Illumina) using patient tumor and blood samples. Alignment to the human reference genome (hg38) and post-alignment analyses were performed using SAREK pipeline[60], as described previously[61]. Median target coverage of tumor samples ranged from 38 to 165 (median = 81) and that of patient-matched normal samples from 37 to 106 (median = 73). Mutations were called using the intersection of VarScan 2.4.2[62] and Strelka2[63] single nucleotide variant (SNV) calls, with default settings for Strelka2 and

filtering of VarScan variants as described previously[61]. Exonic and splice site mutations[64] with a variant allele-frequency >10% were retained. Indels were called using VarScan 2.4.2 as described previously[59]. The final variant set is available as Supplementary Data 1. Loss-of-function mutations were defined as frameshift, nonsense, or splice site mutations or multiple gene events from different categories. *Maftools* oncoplot[65] was used to screen the data for recurrently mutated genes. Mutational contexts were retrieved by *deconstructSigs*[66] with *signatures.cosmic*[67] as reference. Loss-of-Heterozygosity of the *HLA* locus was called visually from plots of B-allele frequencies under the condition of heterozygous germline background. B-allele frequencies of common germline SNPs (dbSNP version 151) were obtained using *samtools* mpileup and *bcftools*[68]. Copy number data were generated using CONTRA 2.0.3[69] and segmented by *GLAD*[70], and were merged with previously obtained copy number data[59], on hg38 co-ordinates. Deep deletions and high amplifications were defined as values <(−1) and >1, respectively. Subclonal mutations were identified using *ABSOLUTE* 1.0.6[71] with settings as described previously[59] and defined as variants with a cancer cell fraction below 0.95 considering a 95% confidence interval. Public mutational data were downloaded from TCGA Pan-Cancer Atlas (gdc.cancer.gov/about-data/publications/pancanatlas), Liu et al.[5] and Riaz et al.[38]. For comparison of mutational burden, SNVs with VAF >= 10% and tumor depth >= 7 reads were retained when such information was available, and each external cohort was combined with our cohort using the set of genes mutated in both datasets. For comparison of mutation landscapes with prior anti-CTLA4 relapse, for the Liu data, mucosal and acral samples and samples taken before/on anti-CTLA4 and on anti-PD1 treatment were removed, "HDEL" and "HIGH_AMP" were used as deep deletions and high amplifications, respectively, and LOH of the HLA locus was visually called from LOH plots of the region. For the Riaz data, single nucleotide variants with VAF >= 10% and tumor depth >= 7 reads from non-acral/mucosal/uveal anti-CTLA4 progressed samples were plotted.

## T cell receptor sequencing

T cell receptor (TCR) sequencing was performed as previously described[35]. Briefly, DNAse I (Thermo Scientific) treated tumor RNA samples were subjected to library preparation using AmpliSeq Immune Repertoire Panel (Illumina) and sequenced using Next-Seq500. Data were analyzed with MiXCR[72,73] and then VDJtools[74] as before[35], with the difference that in the current analysis, low frequency clonotypes were not discarded.

## Bulk RNA sequencing

RNA sequencing data of bulk tumor samples, in part used in a previous study of adoptive T cell therapy[59], were processed to FPKM values using HISAT2 version 2.1.0[75] with hg38 reference and StringTie[76]. Transcripts with the same gene name were summed up, the data were limited to protein-coding genes and log-transformed as $\log_2$(data+1). RNAseq data were quantile-normalized together using *limma*[77]. Six samples without ICB treatment were removed. Principal Component (PC) PC2 values were differentially expressed between a batch variable. PC2 was not associated with immune-, pigmentation- or proliferation-related GO terms, instead with GO terms involving "metabolic process" and "gene expression", which frequently indicate batch effects in in-house datasets. We therefore removed PC2 from the data using *swamp*[78]. Gene signatures[41] were used as mean expression scores of available genes. In addition, a raw count matrix was generated for the same set of samples and genes, where again transcripts with the same gene name were summed up. These data were utilized to test for differential expression of single genes, using DESeq2[79] and adjusting for PC2 values. Gene set enrichment analysis[80] was performed for biological process ontology terms using *clusterProfiler*[81], with 12,513 genes having a standard deviation >0.4 of transformed FPKM values and being ranked by DESeq2 test statistics from raw counts.

## Single cell RNAseq

We generated scRNAseq data from four tumor samples available as finely chopped cryopreserved material. Samples were gently thawed and dissociated using Dri Tumor & Tissue Dissociation Reagent (BD Horizon) according to manufacturer's protocol, with digestion incubation times up to 1 h. Dead cells were removed using Dead Cell Removal Kit (Miltenyi Biotech) prior to processing the remaining single cell suspension using Chromium Single Cell 3' Kit with Dual Index Kit TT Set A sample barcodes (10x Genomics) according to manufacturer's recommendations. Libraries were sequenced on Nova-Seq6000 (Illumina) with read length settings 28-10-10-90 as per 10x Genomics User Guide. The h5 files were processed and merged using the R package Seurat 4.0.1[82], the data were reduced to protein-coding genes, translational (RPS/RPL) and mitochondrial genes (MT-), and genes which a maximum count < = 4 were removed. Cell with less than 500 expressed genes were removed. The data were normalized using SCTransform[83], counts that were zero before transformation were set back to zero, and data was log-transformed as $\log_2$(data+1). This resulted in a dataset of 11,606 genes and 26,053 cells. The data were visualized using UMAP on the top 30 principal components and clusters were identified using FindNeighbors (using top 30 PCs, $k = 15$) and FindClusters (Louvain algorithm, Resolution=0.3) functions of Seurat. Biological identities were assigned to the clusters after manual inspection. B cells were defined as cluster 8 and neighboring cluster 13 cells, without expression of CD3D, CD3E, MITF or SOX10 ($n = 559$). T cells were defined as cluster 5 and 6 cells, without expression of CD79A, MITF and SOX10 ($n = 2,921$). B and T cell data were separately re-normalized using SCTransform, zeros were restored, and data were log-transformed as $\log_2$(data+1). The data were then reduced to curated markers for B and T cells, respectively. The data were visualized using UMAP (without PC reduction) and clusters were identified using FindNeighbors ($k = 10$, cosine distance) and FindClusters (Leiden algorithm, Resolution=0.6). Biological identities were assigned to the clusters after manual inspection. CD8 T cells were defined as either expressing CD8A or CD8B, i.e., having non-zero expression. Similarly, presence of TCF7 or PD1 expression was defined by non-zero expression. The processed bulk and single cell RNAseq gene expression datasets are deposited at GEO with accession number GSE244984.

## Spatial RNA expression

**Nanostring GeoMx.** Region of interests (ROI) from tissue microarray cores were selected using Immunofluorescence with CD3 (T-cell), CD20 (B-cell), PMEL/S100B (tumor cell) and DAPI antibodies. The Cancer Transcriptome Atlas assay was performed on the ROIs according to the manufacturer's instructions (Nanostring, Seattle, WA). Each sample was scaled by a factor to obtain the same 75% quantile, and the data were quantile-normalized and log-transformed as $\log_2$ (data+1). ROIs with limited tumor material in matched IHC cores as well as from patients not included in the study cohort were dismissed. The data were reduced to tumor cell specific genes, using two public single cell RNAseq datasets, GSE115978[15] and GSE120575[25], which were processed as described previously[26], and values > 2 being considered as "expressed". Tumor cell-specific genes were defined as expressed in <20% combined CD4/ CD8 T-cells for both datasets and >10% malignant cells, respectively. To account for varying tumor cell content, tumor ROIs were divided by SOX10 expression. Multiple tumor ROIs of the same patient were merged by mean expression values.

**Visium spatial transcriptomics.** Fresh frozen tumor tissue was sectioned onto Visium Spatial Transcriptomics slides containing 4,992 barcoded spots with 55um diameter. After hematoxylin & eosin staining the sections were permeabilized according to the manufacturer's recommendations, with optimal permeabilization time previously determined to be 24 minutes following Tissue Optimization

Guidelines (10x Genomics). Sequencing libraries were prepared in accordance with Visium Spatial Gene Expression User Guide with Dual Index Kit TT Set A sample barcodes (10x Genomics) according to manufacturer's recommendations. Libraries were sequenced on NovaSeq6000 (Illumina) with read length settings 28-10-10-90 as per 10x Genomics User Guide. Reads were processed together with histology images using SpaceRanger. Data were further processed using Seurat[82]. Specifically for sample MM909_37 a lymph node area was discarded. The samples contained a median of 9,997 median UMI per spot (range 3,987-19,121) and 3,427 median genes per spot (range 1,887-5,497). Protein coding genes were retained and translational genes (RPS, RPL) and mitochondrial genes (MT-) were further removed. Spots with less than 500 expressed genes were removed. The data were merged and normalized using SCTransform[83] (assay=Spatial), counts that were zero before transformation were set back to zero, and data was log-transformed as log2(data+1). Tumor cell-enriched spots were defined as SOX10 expression >=2, and CD3D, CD3E and CD79A expression <=1 (n = 2,766 spots). The curated tumor marker genes were centered and clustered with ConsensusClusterPlus[84], using Euclidean distance and 50 iterations of 80% of spots.

### Multiplex immunofluorescence

Paraffin-embedded TMA core tissue sections (3 μm) were baked for 1 hour at 65 °C and were subjected to deparaffinization and immunofluorescence staining in Roche's automatic samples preparation system (Ventana Discovery Ultra) in the following steps. 1. Deparaffinization in EZ prep (70 °C 8 min). 2. Cell conditioning was applied (CC1, 95 °C 40 min). 3. Blocked with inhibitor CM (37 °C 4 min). 4. Primary antibody incubation. 5. HRP-conjugated antibody incubation (37 °C 16 min) (Roche Discovery OmniMap anti-Rb or anti-Ms HRP). 6.Tyramide-coupled fluorescent dye incubation (37 °C 16 min). 7. Antibody denaturation (CC2, 100 °C 8 min). Steps 4-7 were repeated until all intended markers had been fluorescently labeled. Counterstaining was performed using DAPI (0.75 mM, 37 °C 8 min). Antibodies and fluorophores used in the described staining steps are summarized in Supplementary Tables 2-5.

**Image Acquisition.** All multiplex IF-stained (mIF) TMAs were scanned using the PhenoImager HT (Akoya Biosciences) at 20x magnification. Images were obtained through tile scanning using 7-color whole-slide unmixing filters. These filters included DAPI + Opal 570/690, Opal 480/620/780, and Opal 520. To ensure accurate signal specificity of the obtained images the synthetic Opal library in the image processing software InForm version 2.4.11 (Akoya Biosciences) was used for spectral unmixing. Obtained tiles were subsequently stitched together with QuPath version 0.3.2[85] using a QuPath script available in GitHub.

**Digital image analysis.** QuPath version 0.3.2 was used for all digital image analysis[85]. Visual inspections of the tissue cores were performed to exclude samples of poor-quality including samples with high fluorescent background, insufficient amount of tumor cells and degraded tissue cores. Cell segmentation was performed using the StarDist extension running on the pre-trained model *dsb2018_heavy_augment.pb* in QuPath[85]. The fluorescently labeled markers were analyzed using a machine learning classifier, random forest, trained on multiple measurements in QuPath. After establishing classifiers for each biomarker, they are subsequently combined and applied in a sequential manner. To avoid unexpected classes (e.g. CD20/SOX10, CD20/CD8...), marker calls were assigned to the cells using a pre-specified calling order. The analysis grouped together cells expressing MITF and/or SOX10 as melanoma cells.

Three mIF panels consisting of 6 markers each were evaluated (panel 1: NGFR/MITF/SOX10/CD3/CD20/CD31, panel 2: B2M/MITF/SOX10/CD3/CD20/Ki67, panel 3: CD8/PD1/TCF7/Ki67/FOXP3/SOX10).

Cores with insufficient amount of tumor cells or high background were removed after visual inspection. Using QuPath software, initial marker calls were screened, and a set of manually confirmed calls was used to train marker calling. Final marker calls were assigned to the cells using a pre-specified calling order. B2M, NGFR and MITF were called within SOX10+ cells. TCF7, PD1 and Ki67 were called within CD8+ cells. Percentage of cells with a given call were calculated for each core, as a fraction of all cells of a core, including cells without a call; or as fraction within the relevant cell type (e.g., B2M in SOX10+ cells, or PD1 in CD8+ cells). Additionally, mean MITF intensity was calculated within SOX10+ cells, with MITF intensity defined as log2 (nucleus signal + 1). Multiple cores of the same sample were merged by mean percentage, and MITF intensity was merged by mean intensity. Panels were combined using NGFR in SOX10+ cells, CD3 and CD20 from panel 1, MITF/B2M in SOX10+ cells and MITF intensity in SOX10+ cells from panel 2, and CD8 and TCF7/PD1/Ki67 in CD8+ cells from panel 3 (Supplementary Tables 2-5).

### Statistical analyses

Bioinformatical analyses were performed using R version 4.0.5. For group comparisons, T-test and Wilcoxon test were used for two groups, Anova and Kruskal-Wallis test for more than two groups. Pearson correlation was used to compare numerical variables, and Fisher's exact test for categorical variables. All tests were two-sided. False discovery rate was calculated using Benjamini-Hochberg adjustment. Boxplots are displayed with the center-line as median, the box limits as lower and upper quartiles, and with whiskers covering the most extreme values within 1.5 x Interquartile-Range.

### Reporting summary

Further information on research design is available in the Nature Portfolio Reporting Summary linked to this article.

## Data availability

Public mutational data was downloaded from TCGA Pan-Cancer Atlas (gdc.cancer.gov/about-data/publications/pancanatlas), Liu et al.[5] and Riaz et al.[38]. Gene signatures were obtained from referenced publications, respectively. Publicly available data with accession numbers GSE115978 and GSE120575 were downloaded from Gene Expression Omnibus (GEO) and were used to identify tumor-specific genes.

Processed bulk RNA sequencing and single cell RNA sequencing data have been deposited at GEO with accession number GSE244982 and GSE244983. Raw data are not available for these GEO submissions, as due to Swedish and Danish laws, the patient consent, and the risk that the sequencing data contains personally-identifiable information and hereditary mutations, we cannot deposit the short sequencing read data in a public access repository.

Spatial transcriptomics data have been deposited under accession number GSE261347.

Whole exome sequencing- and T cell receptor sequencing data were deposited in European Genome Archive (EGA) under EGAD50000000380 and EGAD50000000379, respectively. These data are available under restricted access. Data access can be granted via the EGA under collaborative conditions and when aligned with current ethical approval, and data will be available for duration of the proposed project. Somatically called mutations are available as Supplementary Data 1.

All other remaining data are available within the Article, Supplementary Information or as Source data file. Source data are provided with this paper.

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

## Acknowledgements

This work was supported by Swedish Research Council (Vetenskapsrådet Dnr 2018-02786, Dnr 2022-00871, GJ), Swedish Cancer Society (19 0458 Pj, 22 2105 Pj, GJ), Berta Kamprad Foundation (GJ and BP) and the governmental funding for healthcare research (ALF and GJ), Knut and Alice Wallenberg Foundation (KAW 2022.0066, GJ) and Göran Gustafsson Foundation (GJ). The authors would like to acknowledge Clinical Genomics Lund, SciLifeLab and Center for Translational Genomics (CTG), Lund University, for providing expertise and service with sequencing and analysis. The computations and data handling were enabled by resources provided by the Swedish National Infrastructure for Computing (SNIC) at Uppsala Multidisciplinary Center for Advanced Computational Science (UPPMAX) partially funded by the Swedish Research Council through grant agreement no. 2018-05973. We also thank Nanostring for GeoMx experimental analyses.

## Author contributions

M.L. conducted all data analysis, design of study and writing the manuscript. B.P. conducted all immunostaining and analysis of such data.

T.H.B. retrieved and interpreted clinical data from immunotherapy resistant cases. K.H. conducted whole-exome-, RNA-, TCR and single cell RNA sequencing. K.K. performed single cell RNA sequencing. A.E. and I.H. performed sectioning for Visium sequencing. J.Y. conducted and analysed single cell RNA sequencing data from B cells. K.N., C.I., A.C. and K.I. collected clinical information on immune checkpoint blockade naïve patients and set up protocols for collection of viable tumor tissue from patients with melanoma. K.P. performed and analysed multiplex immunofluorescence data. I.M.S. designed study, collected and interpreted clinical data from immunotherapy resistant cases. M.D. supervised and designed study, collected and interpreted clinical data from immunotherapy resistant cases. G.J. supervised and designed study and wrote the manuscript. All authors have read and approved the manuscript.

## Funding

## Competing interests
The authors have no competing interests.
