## [Peer Review File · Nature Communications]

Molecular patterns of resistance to immune checkpoint blockade in melanomaREVIEWER COMMENTS

Reviewer #1 (Remarks to the Author): with expertise in melanoma, (immuno)therapy

I would like to congratulate the authors for the comprehensive analysis of a largish cohort of ICB resistant melanoma. The manuscript is well written, although the figures/tables and some areas of text would benefit from improved clarity. I have itemised several examples for the authors. I also felt that the authors did not adequately acknowledge previous literature - even though some of these previous studies are not in resistant ICB samples, the role of antigen presentation and dedifferentiation have been highlighted several times and should be cited. The biggest issue for this reviewer is how this work adds to what we already know. The authors report some interesting associations, but no new mechanisms of ICB resistance are identified or validated. In some cases the reported findings are also not supported by the data, and the numbers of samples used to make conclusions become increasingly small. For instance, the data in Figure 2C does not show CD3 higher in CTLA4res as indicated in the text - the p value is 0.12, and it is clear that most samples overlap.

Specific comments

Results, paragraph 1: The sentence describing 12 anti-PD1 resistant samples taken at day 7 during BRAFi treatment is not clear based on Table 1. These 12 patients are listed as previous BRAFi by protocol - the results text reads as if PD1 preceded BRAF inhibitor therapy?

Overall the figures are often difficult to read (text is very small in many cases: example Figure S2), and the legends are not detailed enough

The separation of mucosal is not clearly described in the text, but appears in Table 1 - this needs to be detailed more carefully- are the authors assuming mucosal have alternate mechanisms of resistance to cutaneous melanoma?

In identifying genetic resistance, why was $n \geq 4$ selected as high frequency cut off for mutant genes in ICB resistance - and it is not clear if the authors restricted this to the recurrent hot spot mutations mutations. The comparison with non ICB melanoma has been

done apparently - this is mentioned in the results - but I don't see the data for this comparison i.e. relative to ICB naive melanoma in Figure S2. The authors should also include analysis of previously submitted ICB samples - there are resistant data published (Zaretsky NEJM 2016; Shin 2017 Cancer Dis, Sade-Feldman 2017 Nature Com, Lim et al. 2023 Nature Comm; and consider PRE-treatment samples could act as a nice control for identifying resistance genes; Newell Cancer Cell 2022).

The analysis of HLA-I highlights the limitation of the previous genome wide analysis and $n \geq 4$ gene mutation cut off - better to select a signaling pathway and examine enrichment of alterations - for instance - when looking at HLA-I presentation - the alterations include 3 B2M changes, and 3 LOH HLA. Individually this doesn't meet the arbitrary $n \geq 4$, but as a process you have 6 tumours with alterations.

Table 1 is not as clear as needed - define CTLA4 res and PD1 res - the resistance types 'probably primary etc' need to be defined as a footnote and in the text - there is no mention of resistance definition and what probably means in this instance? the numbers are also confusing - presumably because mucosal is included in Table 1 - but not included in the initial Results details - i.e. 23 mets prog on CTLA4; IL2 details not in Table 1

Reviewer #2 (Remarks to the Author): with expertise in melanoma, (immuno)therapy

This is an interesting article describing analyses of biopsies of patients with advanced melanoma upon progression after treatment with anti-CTLA-4 or anti-PD-1.

Major comments:

It would have been desirable to include comparison with baseline biopsies prior to any immune checkpoint blockade (ICB) therapy; it is acknowledged that they may not be available for the majority of cases, but it is hard to think that there were none for comparison as it is routine to biopsy first sites of metastases before starting on therapy.

The authors should specify if the deep deletions in B2M or JAK2 were considered to result in

loss of both alleles of the gene. Were any of these cases stained for B2M to confirm that the protein was not present in the melanoma cells?

Many of the conclusions of different paragraphs include sentences that are based on associations but are presented as being causative events. For example, at the bottom of page 5 “This suggests that anti-PD1 resistance may lead to downregulation of melanocytic antigens that subsequently leads to immune escape”, but based on the data presented, it is unclear that the lack of melanocytic antigens is the cause or a consequence of PD-1 blockade resistance.

Also, at the top of page 6 “In summary, the observations indicate a sustained immune response in some tumors despite progressing on anti-CTLA4; in contrast, a particularly immune-poor microenvironment was observed in anti-PD1 resistant melanoma” is based on data that could be interpreted differently, where some immune-rich cases in the anti-CTLA-4 cohort are not represented in the anti-PD-1 cohort because anti-PD-1 has a higher response rate, and those cases may have an antitumor response limited by the PD-1 checkpoint and not the CTLA-4 checkpoint.

Furthermore, at the top of page 7 “In conclusion, melanoma metastases harbor multiple tumor cell populations that are correlated to T cell infiltration” should be modified, as this article provides, at best, loose associations between descriptive analyses and patient outcomes, as opposed to true “correlations”.

The Discussion should also acknowledge that lack of immunogenicity of the cancer cells would lead to lack of response to ICB therapies, as the immune system needs to differentiate between normal and malignant cells to result in clinical responses. Therefore, some of the cases may have no other cancer cell-intrinsic alteration other than low immunogenicity.

Minor comments:

It would be desirable to make the patient the noun, as opposed to being an adjective to

their cancer (write “patients with metastatic melanoma” as opposed to “metastatic melanoma patients”).

What makes the cohort “unique”, as stated in the Abstract? Would be best to just delete this word, which adds no real meaning to the sentence.

Reviewer #3 (Remarks to the Author): with expertise in computational, melanoma, (immuno)therapy

In this study, Lauss et al. use newly collected biopsies from melanoma patients resistant to aCTLA4/aPD1 treatment, to examine the genetic and molecular landscape associated with treatment resistance. Using newly collected samples they perform WES, bulk and scRNAseq as well as spatial transcriptomics to provide a comprehensive dataset for investigation of different aspects of immunotherapy relapse/resistance. By comparing mutations and gene expression patterns to baseline from publicly available datasets, the authors pinpoint key genomic and molecular patterns that characterize resistance to aCTLA4 vs aPD1.

This study addresses an important objective and provides a substantial amount of new data that would be highly useful to the scientific community when made available. In addition to providing the baseline data, there are several aspects of the computational and/or statistical approaches that must be revised to ensure correctness of the comprehensive analysis in this study.

1. The data must be made available. First, the authors should deposit the data into GEO and provide the accession (it can be made available at publication but deposited in advance). In addition to the bulk RNAseq, the authors must provide the processed mutation data in supp. Info, and additionally deposit the processed scRNAseq to GEO.
2. 17 of the 21 patients in the aPD1 group relapsed or progressed on prior CTLA4. Therefore, the differences are not actually between patients resistant to aCTLA4 and those resistant to aPD1, because the majority in both groups are patients resistant to aCTLA4. This should be addressed or rephrased.
3. In absence of paired pre and at-relapse biopsies, the authors use the TCGA as proxy for

treatment naïve samples. There are two issues that should be addressed. First, it is unclear what statistical test was applied for the authors to conclude that “... this frequency was not different as compared to the treatment-naïve distant metastases from TCGA”? They should support this using a Chi squared test and provide the p-value for every gene examined.

Second, the authors should use patients that relapse with aCTLA4 as baseline for the aPD1 resistant analysis (available from Riaz/Liu, which they analyzed in supp.).

4. The use of Cufflinks and Tophat2 is inappropriate, as Tophat2 is deprecated: “Please note that TopHat has entered a low maintenance, low support stage as it is now largely superseded by HISAT2 which provides the same core functionality (i.e. spliced alignment of RNA-Seq reads), in a more accurate and much more efficient way”:

<http://ccb.jhu.edu/software/tophat/index.shtml>

The authors should revise the pre-processing and use an updated software (e.g. STAR-RSEM)

5. The authors seem to use t-test for differential expression analysis of RNA sequencing, which is also inappropriate, as the read counts follow a negative binomial distribution, e.g.:

<https://genomebiology.biomedcentral.com/articles/10.1186/gb-2010-11-10-r106>

Which is why DESEQ2 or a similar software developed for RNAseq should be applied.

6. There are some mislabeled figures (e.g. Fig2), missing legends (e.g. Fig3A), typos (e.g. use of “thereof”), that should be corrected.

7. The following conclusion “In conclusion, our work provides a comprehensive view of molecular and immune cell changes occurring at relapse on ICB. “ is not correct as stated, because there is no support that any of these changes occurred at relapse, when the baseline is treatment naïve samples.

REVIEWER COMMENTS

Reviewer #1 (Remarks to the Author)

I would like to congratulate the authors for the comprehensive analysis of a largish cohort of ICB resistant melanoma. The manuscript is well written, although the figures/tables and some areas of text would benefit from improved clarity. I have itemised several examples for the authors. I also felt that the authors did not adequately acknowledge previous literature - even though some of these previous studies are not in resistant ICB samples, the role of antigen presentation and dedifferentiation have been highlighted several times and should be cited. The biggest issue for this reviewer is how this work adds to what we already know. The authors report some interesting associations, but no new mechanisms of ICB resistance are identified or validated. In some cases the reported findings are also not supported by the data, and the numbers of samples used to make conclusions become increasingly small. For instance, the data in Figure 2C does not show CD3 higher in CTLA4res as indicated in the text - the p value is 0.12, and it is clear that most samples overlap.

- **Reply:** We thank the reviewer for the overall positive response. Importantly, our study of progressed samples describes specific tumor microenvironments that are most likely molded from immunotherapy resistance which adds novelty to the field. Moreover, the anti-PD1 resistant tumors have an immune-poor microenvironment, whereas tumor that solely progressed on anti-CTLA4 had different composition of immune cells in the microenvironment. These findings may have important consequences for further treatment and treatment sequence of ICB progressed patients. The reviewer is also correct that sample size issues occur, specifically when analyzing patient subsets. Please note that studies on progressed samples are rare and small in sample size, and for melanoma range from a few samples to up to 22 (which were patient derived cell lines from resistant patients). We have now summarized these studies in the Discussion section. In this regard our sample size is considerable and should have allowed us to find resistance features with large effects. To find resistance features with relatively small effects, still larger sample sizes of progressed patients will be needed. Finally, using immunofluorescence images, several anti-CTLA4 resistant cases showed high fraction of CD3+ T cells, in line with RNAseq findings. However, as noted by the reviewer, we now clarify that on the immunofluorescence level the difference to anti-PD1 resistant tumors was not statistically significant, page 5.

Specific comments

Results, paragraph 1: The sentence describing 12 anti-PD1 resistant samples taken at day 7 during BRAFi treatment is not clear based on Table 1. These 12 patients are listed as previous BRAFi by protocol - the results text reads as if PD1 preceded BRAF inhibitor therapy?

- **Reply:** We thank the reviewer and apologize for this confusion. These patients had relapsed on anti-PD1 therapy, and then entered a trial using BRAFi. Biopsies were taken at day 7 by protocol. We have now clarified this in Table 1 and in the text on page 3.

Overall the figures are often difficult to read (text is very small in many cases: example Figure S2), and the legends are not detailed enough

- **Reply:** We agree with the reviewer that some of the Figures were a bit unclear. We have now revised and improved Figures and Figure Legends.

The separation of mucosal is not clearly described in the text, but appears in Table 1 - this needs to be detailed more carefully- are the authors assuming mucosal have alternate mechanisms of resistance to cutaneous melanoma?

- **Reply:** We thank the reviewer for highlighting this and agree that this needs to be clarified in the text. We have treated the mucosal melanomas separately as we know that such melanomas harbor distinct biological characteristics different from cutaneous melanomas (Newell et al. Can Discovery 2022). We also know that mucosal melanoma patients have very limited response to PD1 and CTLA4 blockade (Weber et al. NEJM 2017). We show basic genomic and transcriptomic data of mucosal samples but have excluded them from downstream statistical analyses. In the revised text, we have now motivated excluding the mucosal samples for downstream statistical analyses, on page 3.

In identifying genetic resistance, why was $n \geq 4$ selected as high frequency cut off for mutant genes in ICB resistance - and it is not clear if the authors restricted this to the recurrent hot spot mutations mutations. The comparison with non ICB melanoma has been done apparently - this is mentioned in the results - but I dont see the data for this comparison i.e relative to ICB naive melanoma in Figure S2. The authors should also include analysis of previously submitted ICB samples - there are resistant data published (Zaretsky NEJM 2016; Shin 2017 Cancer Dis, Sade-Feldman 2017 Nature Communications, Lim et al. 2023 Nature Communications; and consider PRE-treatment samples could act as a nice control for identifying resistance genes; Newell et al. Cancer Cell 2022).

- **Reply:** We appreciate the concern from the reviewer on the mutation cutoff. The cutoff of $n \geq 4$ was not applied to known melanoma driver genes and to genes reportedly involved in ICB resistance. Instead, the cutoff was applied in the whole-genome discovery of hot-spot mutations and loss-of function mutations, with the aim to reduce false positive findings. To illustrate this, reducing the cutoff to e.g., ≥ 2 mutations, it becomes immediately clear that the results are littered with false positive results (Fig R1.1).

Fig R1.1. Recurrent hotspot mutations, i.e. affecting the same amino acid, excluding silent mutations, in at least two ICB resistant melanoma from a genome-wide analysis. Affected amino acid number is indicated after the gene name.

As we used a strict cutoff of ≥ 4 in the main text, Supplementary Figure 2 C-D is actually based on a cutoff of ≥ 3 , that includes the known cancer driver genes *APC* and *PCLB4*.

We have also added a comparison of ICB resistant to ICB naïve samples based on Fisher's exact test to Fig. S1B of the revised manuscript.

We highly welcome the reviewer's suggestion to include published findings in the manuscript. We have added a paragraph summarizing these publications in the Discussion section:

"Specifically, a landmark study reported *JAK1*, *JAK2* and *B2M* LoF mutations in three of four investigated ICB resistant samples, respectively (Zaretsky et al. NEJM 2016). Another study highlighted 5 of 12 progressing patients to harbor *B2M* LoF mutations or LOH (Sade-Feldman et al. Nature Communications 2017). In a recent study, 22 cell lines from 18 patients that had progressed on anti-PD1 or anti-PD1/anti-CTLA4 therapy, contained one *JAK2* and two *B2M* inactivating events and two *HLA* LOH events, next to other potential ICB resistance mechanisms (Lim et al. Nature Communications 2023)".

The study from Newell *et al* (Cancer Cell 2022), reports tumor mutational burden and an Interferon-gamma signature as the best bivariable model of response in 77 ICB baseline samples. We have added this reference at the relevant section on baseline predictors in the Introduction section. The study from Shin *et al* (Can Discovery 2017) also uses baseline samples and is already referred to in the Introduction section.

The analysis of HLA-I highlights the limitation of the previous genome wide analysis and $n \geq 4$ gene mutation cut off - better to select a signaling pathway and examine enrichment of alterations - for instance - when looking at HLA-I presentation - the alterations include 3 B2M changes, and 3 LOH HLA. Individually this doesn't meet the arbitrary $n \geq 4$, but as a process you have 6 tumours with alterations.

- **Reply:** We completely agree with the reviewer and did not include any cutoff when it comes to the known resistance and immune evasion signaling pathways. The cutoff was necessary to reduce false positives in genome-wide discovery analyses, please see previous comment.

Table 1 is not as clear as needed - define CTLA4 res and PD1 res - the resistance types 'probably primary etc' need to be defined as a footnotes and in the text - there is no mention of resistance definition and what probably means in this instance? the numbers are also confusing - presumably because mucosal is included in Table 1 - but not included in the initial Results details - i.e 23 mets prog on CTLA4; IL2 details not in Table 1

- **Reply:** We thank the reviewer for highlighting this and have clarified the Table considerably, included footnotes and have synchronized Table 1 with the text. In contrast to e.g., lung cancer (Schoenfeld et al. Ann. Oncol. 2021), a consensus on what comprises acquired resistance in melanoma has not been reached yet, particularly in the setting of stable disease. To avoid confusion, we have combined "probably primary" and "probably acquired" with "primary" and "acquired" resistance, respectively. We have clarified exclusion of mucosal melanoma from downstream statistical analyses in the revised text, and added information on previous IL2 treatment to Table 1.

Reviewer #2 (Remarks to the Authors)

This is an interesting article describing analyses of biopsies of patients with advanced melanoma upon progression after treatment with anti-CTLA-4 or anti-PD-1.

Major comments:

It would have been desirable to include comparison with baseline biopsies prior to any immune checkpoint blockade (ICB) therapy; it is acknowledged that they may not be available for the majority of cases, but it is hard to think that there were none for comparison as it is routine to biopsy first sites of metastases before starting on therapy.

- **Reply:** We agree with the reviewer that this indeed would have been desirable. However, despite efforts we have not been able to get hold of pre-treatment biopsies. Much of this is due to tissue blocks being stored in central repositories across Denmark after a certain time from diagnosis. Withdrawal of tissue blocks for research purposes from such repositories is a long process and would have delayed the manuscript considerably. In addition, pre-treatment core biopsies from metastatic tissue are available only from a minority of cases. However, we do not believe that this will change the conclusions drawn from this study. Nevertheless, in the revised manuscript we have added comparisons on the genomic level to baseline samples from Liu *et al.* (Nature Medicine 2019) and Riaz *et al.* (Cell 2017) cohorts (Fig. S1D). Neither to these samples nor to ICB naïve TCGA samples did we identify substantial genomic differences, which may be attributable to the majority of our patients having primary resistance to ICB.

The authors should specify if the deep deletions in B2M or JAK2 were considered to result in loss of both alleles of the gene. Were any of these cases stained for B2M to confirm that the protein was not present in the melanoma cells?

- **Reply:** We thank the reviewer for these observations. We found two deep deletions of B2M and further analysis of the sequencing data suggests that in both cases both alleles were lost. One of the cases were stained for B2M in a multiplex immunofluorescence panel (DAPI, SOX10, MITF, B2M, CD3, CD20) and demonstrated obvious loss of the B2M protein in the tumor cells (Fig. R2.1). We have added this example to the revised manuscript as Supplementary Figure 3.

Supplementary Figure 3

Fig R2.1: Deep deletion of B2M. Upper panel: Genome-wide copy number profile of Pat49 with a zoom-in of the B2M region +/- 8 Megabases reveals a focal deep deletion of B2M. red points = segmented values. Lower panel: Multiplex-immunofluorescence of Pat49 and positive control confirms that tumors cells of Pat49 lack expression of B2M.

Many of the conclusions of different paragraphs include sentences that are based on associations but are presented as being causative events. For example, at the bottom of page 5 “This suggests that anti-PD1 resistance may lead to downregulation of melanocytic antigens that subsequently leads to immune escape”, but based on the data presented, it is unclear that the lack of melanocytic antigens is the cause or a consequence of PD-1 blockade resistance.

- **Reply:** We understand and agree to the comment from the reviewer and subsequently have gone through the text. The following sentences were changed - “This suggests that anti-PD1 resistance may lead to downregulation of melanocytic antigens that subsequently leads to immune escape” was changed to “This suggests that anti-PD1 resistance correlates with decreased melanocytic antigens which may facilitate immune escape.”

“Thus, TCR clonality analysis suggests that BRAFi induces an influx of T cells in melanomas but not expansion of specific TCR clones.” Was changed to “Thus, TCR clonality analysis suggests that BRAFi is associated with a higher abundance of T cells in melanomas but not expansion of specific TCR clones.”

“In conclusion, our work provides a comprehensive view of molecular and immune cell changes occurring at relapse on ICB. “ in line with reviewer 3 was changed to “In conclusion, our work provides a comprehensive view of the molecular and immune cell landscape at relapse to ICB”

Also, at the top of page 6 “In summary, the observations indicate a sustained immune response in some tumors despite progressing on anti-CTLA4; in contrast, a particularly immune-poor microenvironment was observed in anti-PD1 resistant melanoma” is based on

data that could be interpreted differently, where some immune-rich cases in the anti-CTLA-4 cohort are not represented in the anti-PD-1 cohort because anti-PD-1 has a higher response rate, and those cases may have an antitumor response limited by the PD-1 checkpoint and not the CTLA-4 checkpoint.

- **Reply:** This is a valid point from the reviewer and we have included this in the discussion section. We changed the sentence:
“Overall, this suggests that the tumor specific immune response is considerably hampered when anti-PD1 resistance is developed.” to
“Overall, this suggests that the tumor specific immune response is considerably hampered in PD1 resistant cases, which may either have developed during resistance or has pre-existed and was selected for due to the high response rate of PD1 blockade.”

Furthermore, at the top of page 7 “In conclusion, melanoma metastases harbor multiple tumor cell populations that are correlated to T cell infiltration” should be modified, as this article provides, at best, loose associations between descriptive analyses and patient outcomes, as opposed to true “correlations”.

- **Reply:** We have now toned down this part of the manuscript and changed the wording from “correlation” to “association”.

The Discussion should also acknowledge that lack of immunogenicity of the cancer cells would lead to lack of response to ICB therapies, as the immune system needs to differentiate between normal and malignant cells to result in clinical responses. Therefore, some of the cases may have no other cancer cell-intrinsic alteration other than low immunogenicity.

- **Reply:** Yes, we agree with the reviewer that this might be the case. In principle, melanoma tumors should have elevated immunogenicity due to one of the highest tumor mutational burdens across cancer types (Alexandrov et al. Nature 2013) and due to many of the melanocytic genes being self-antigens targeted by the immune system. Nevertheless, it is possible for tumors to evade the immune system through various mechanism, including reduced, e.g., reduced neoantigen presentation, or down-regulation of melanocyte differentiation antigens. Additionally, we have focused on the tumor microenvironment of ICB resistant melanomas. We have now discussed the issue of low immunogenicity in the revised Discussion section.

Minor comments:

It would be desirable to make the patient the noun, as opposed to being an adjective to their cancer (write “patients with metastatic melanoma” as opposed to “metastatic melanoma patients”).

- **Reply:** This has now been changed in the revised manuscript.

What makes the cohort “unique”, as stated in the Abstract? Would be best to just delete this word, which adds no real meaning to the sentence.

- **Reply:** We have now removed “unique” from the Abstract.

Reviewer #3 (Remarks to the Author)

In this study, Lauss et al. use newly collected biopsies from melanoma patients resistant to aCTLA4/aPD1 treatment, to examine the genetic and molecular landscape associated with treatment resistance. Using newly collected samples they perform WES, bulk and scRNAseq as well as spatial transcriptomics to provide a comprehensive dataset for investigation of different aspects of immunotherapy relapse/resistance. By comparing mutations and gene

expression patterns to baseline from publicly available datasets, the authors pinpoint key genomic and molecular patterns that characterize resistance to aCTLA4 vs aPD1. This study addresses an important objective and provides a substantial amount of new data that would be highly useful to the scientific community when made available. In addition to providing the baseline data, there are several aspects of the computational and/or statistical approaches that must be revised to ensure correctness of the comprehensive analysis in this study.

1. The data must be made available. First, the authors should deposit the data into GEO and provide the accession (it can be made available at publication but deposited in advance). In addition to the bulk RNAseq, the authors must provide the processed mutation data in supp. Info, and additionally deposit the processed scRNAseq to GEO.

- **Reply:** We thank the reviewer for addressing this issue. We have now deposited the bulk RNAseq and single cell RNAseq data to GEO with accession number GSE244984 (GSE244982 and GSE244983, respectively). The reviewer token for access to the data is irihkemctnadlgb Mutation data have been added as Supplementary Data 1.

2. 17 of the 21 patients in the aPD1 group relapsed or progressed on prior CTLA4. Therefore, the differences are not actually between patients resistant to aCTLA4 and those resistant to aPD1, because the majority in both groups are patients resistant to aCTLA4. This should be addressed or rephrased.

- **Reply:** This is a correct point by the reviewer. We further clarified this in the beginning of the Results section, and in addition have addressed this in the Discussion section of the revised manuscript.

3. In absence of paired pre and at-relapse biopsies, the authors use the TCGA as proxy for treatment naïve samples. There are two issues that should be addressed. First, it is unclear what statistical test was applied for the authors to conclude that "... this frequency was not different as compared to the treatment-naïve distant metastases from TCGA"? They should support this using a Chi squared test and provide the p-value for every gene examined. Second, the authors should use patients that relapse with aCTLA4 as baseline for the aPD1 resistant analysis (available from Riaz/Liu, which they analyzed in supp.).

- **Reply:** We thank the reviewer for this observation and have added p-values from Fisher's test for each gene to the comparison of our data to TCGA data (Fig. S1B of the revised manuscript). We also agree with the reviewer that anti-PD1 resistant tumors with prior relapse to CTLA4 should be compared with publicly available anti-CTLA4 relapsed samples as baseline. Therefore, we extracted such data from Liu *et al* (Nat Med 2019) and Riaz *et al* (Cell 2017) publications and added this comparison as Fig. S1D (Fig. R3.1). For Liu *et al* (Nat Med 2019), where information on mutation-, copy number- and LOH level was available, we also added p-values from Fisher's test in comparison to our data (Fig. R3.1). In the main text we concluded: "Further, we compared anti-PD1 resistant patients with prior relapse to anti-CTLA4, to anti-PD1 naïve samples with prior relapse to anti-CTLA4 from two public datasets (Liu et al. Nat Med 2019, Riaz et al. Cell 2017) to isolate the effect of PD1 blockade. However, we found similar mutational landscapes across the datasets (Fig. S1D)." The Methods section has been updated accordingly.

Fig R3.1 Genetic aberrations of selected genes in anti-PD1 resistant samples with prior relapse to CTLA4 blockade, in comparison to publicly available anti-CTLA4 relapsed samples from Liu et al. and Riaz et al.. Frequency plots of activating events for potential oncogenes and loss-of-function events for potential tumor suppressor genes are depicted for ICB resistant tumors and for the Liu et al. control cohort, respectively, P-values from Fisher test. For Riaz et al, only mutational data were available.

4. The use of Cufflinks and Tophat2 is inappropriate, as Tophat2 is deprecated: “Please note that TopHat has entered a low maintenance, low support stage as it is now largely superseded by HISAT2 which provides the same core functionality (i.e. spliced alignment of RNA-Seq reads), in a more accurate and much more efficient way”: <http://ccb.jhu.edu/software/tophat/index.shtml>
The authors should revise the pre-processing and use an updated software (e.g. STAR-RSEM)

- **Reply:** The bulk RNAseq processing and analyses have now been redone with HISAT2 and as expected the results are very similar, however with more p-values crossing the significance threshold. Figure 2A-B and Fig. S4 have been replaced with revised versions. The Methods section has been updated accordingly.

5. The authors seem to use t-test for differential expression analysis of RNA sequencing, which is also inappropriate, as the read counts follow a negative binomial distribution, e.g.: <https://genomebiology.biomedcentral.com/articles/10.1186/gb-2010-11-10-r106>
Which is why DESEQ2 or a similar software developed for RNAseq should be applied.

- **Reply:** Please note, that we do not use count data directly, instead we use FPKM values to reduce library size effects. These FPKM values were further log-transformed as $\log_2(\text{data}+1)$. Conveniently, these transformed data have a stable variance from lowly to highly expressed genes (Fig. R3.2, left panel). More importantly, gene expression of the normalized data is roughly log-normal, i.e. the genes follow a normal distribution (Fig. R3.2, right panel), which allows for the use of t-test.

Fig R3.2: Properties of log-transformed FPKM values of the present bulk RNAseq data. Left panel: Relationship of mean to variance for each gene. Variance is in a similar range for lowly and highly expressed genes. Red= lowest smoothed line. Right panel: Distribution of p-values obtained from Shapiro-Wilk test for normal distribution. Normal distribution was not rejected ($p<0.05$) for 8,945 (50%) of 17,896 genes with non-zero variation.

6. There are some mislabeled figures (e.g. Fig2), missing legends (e.g. Fig3A), typos (e.g. use of “thereof”), that should be corrected.
- **Reply:** We have now revised and improved Figures and Figure Legends, and the use of “thereof” was corrected.
7. The following conclusion “In conclusion, our work provides a comprehensive view of molecular and immune cell changes occurring at relapse on ICB. “ is not correct as stated, because there is no support that any of these changes occurred at relapse, when the baseline is treatment naïve samples.
- **Reply:** We agree with the reviewer and have changed the wording of the statement to. “ In conclusion, our work provides a comprehensive view of the molecular and immune cell landscape at relapse to ICB.”

REVIEWER COMMENTS

Reviewer #1 (Remarks to the Author):

The authors have certainly improved the manuscript, but clarity in the text and figure legends remains an issue.

My comment on Results, paragraph 1 has not been adequately addressed. It would be best not to have these 12 anti-PD1 resistant patients under the subheading 'Previous BRAFi' in Table 1?

I am still struggling with some figures and legends- Figure S2 is a good example - why are the genes repeated in the rows, I couldnt see an arrowhead in TP53?

Figure S4 - what is the anti-PD1* group in this figure. Figure S5 - I am not sure how we can decipher these images ? Figure S1B needs some work - the labels on the TMB graphs are unclear - these are resistant lesions from melanoma patients - and the response data for each patients is provided? then what are CTLA4res lesions? The authors need to review carefully, making sure that each figures is well described, and consistent labels are used throughout.

Reviewer #2 (Remarks to the Author):

The authors have correctly addressed the comments from the reviewers that they were able to incorporate in a timely manner.

Reviewer #3 (Remarks to the Author):

The authors have comprehensively addressed almost all our concerns. The paper reports important findings/data and should be published. However, one concern is remaining, which should be addressed before this paper is ready for publication. That is, the test used for differential expression analysis. We are not aware of a protocol for applying t-test to the log transformed FPKM, or to any transformation to the FPKM values (e.g. see potential issues in [1]). Conventional test that for differential expression (DESEQ2 or edgeR) should be

applied unless there is a strong reasoning against their validity, which we could not find in this case.

In their response, the authors show that only a subset of the genes follows the described edge case that they claim support application of a t-test. Do all of the genes assigned with a significant p-value by t-test pass the Shapiro-Wilk test?

The authors should, at the very least, include results in supplementary where DESEQ2 is applied to the read counts to yield differentially expressed genes, showing that their results are similar.

[1] <https://www.nature.com/articles/nbt.3682>

REVIEWER COMMENTS

Reviewer #1 (Remarks to the Author):

The authors have certainly improved the manuscript, but clarity in the text and figure legends remains an issue.

My comment on Results, paragraph 1 has not been adequately addressed. It would be best not to have these 12 anti-PD1 resistant patients under the subheading 'Previous BRAFi' in Table 1?

I am still struggling with some figures and legends- Figure S2 is a good example - why are the genes repeated in the rows, I couldnt see an arrowhead in TP53?

Figure S4 - what is the anti-PD1* group in this figure. Figure S5 - I am not sure how we can decipher these images ? Figure S1B needs some work - the labels on the TMB graphs are unclear - these are resistant lesions from melanoma patients - and the response data for each patients is provided? then what are CTLA4res lesions? The authors need to review carefully, making sure that each figures is well described, and consistent labels are used throughout.

Reply: *We appreciate the concerns from the reviewer and agree and as such we have clarified some items. In the text we have now used the same abbreviations as we use in the figure legends and figures to make it easier to follow. We have also expanded and improved all figure legends as well as making sure there is consistency between all figures.*

Reviewer #3 (Remarks to the Author):

The authors have comprehensively addressed almost all our concerns. The paper reports important findings/data and should be published. However, one concern is remaining, which should be addressed before this paper is ready for publication. That is, the test used for differential expression analysis. We are not aware of a protocol for applying t-test to the log transformed FPKM, or to any transformation to the FPKM values (e.g. see potential issues in [1]). Conventional test that for differential expression (DESeq2 or edgeR) should be applied unless there is a strong reasoning against their validity, which we could not find in this case.

In their response, the authors show that only a subset of the genes follows the described edge case that they claim support application of a t-test. Do all of the genes assigned with a significant p-value by t-test pass the Shapiro-Wilk test?

The authors should, at the very least, include results in supplementary where DESeq2 is applied to the read counts to yield differentially expressed genes, showing that their results are similar.

[1] <https://www.nature.com/articles/nbt.3682>

Reply: *We appreciate the reviewer's comment regarding differential expression analysis. We have now used DESeq2 on raw counts to compare with t-test on transformed FPKM values. As we removed Principal Component 2 (PC2) of the data due to technical effects, we therefore also adjusted DESeq2 for*

PC2 values. Testing for differences between anti-CTLA4 resistant and anti-PD1 resistant cases, the correlation of test statistics between DESeq2 and t-test was 0.79, and the correlation of $-\log_{10}$ p-values was 0.57 (Fig R3.1). To disregard the effect of PC2 correction, we tested the uncorrected data for 50 randomly obtained group variables. The correlation between DESeq2 and t-test statistics was high, with a median Pearson correlation of 0.83 (Fig R3.2). Biological processes and cellular composition that are underlying gene expression, give rise to a variety of distributions, such as bimodal, multi-modal, uniform distributions etc., which are distinct from normal or negative binomial distributions. Since 50% of the genes were normally distributed in the transformed FPKM data, we would argue that these data are roughly log-normal. However, the assumption of negative binomial distribution is theoretically appealing, and we have now included p-values from DESeq2 on raw counts in FigS4B. The majority of tested immune genes had even lower p-values using DESeq2 than using t-test (Fig R3.3), confirming our biological conclusions. We repeated the gene set enrichment analysis of Fig 2B with genes ranked by DESeq2 test statistics and received almost identical top-scoring pathways as compared with t-test (Fig R3.4). In the revised manuscript, we now use the results from DESeq2 in Fig 2B. Overall, the results using t-test on transformed FPKM values and DESeq2 on raw counts were similar. We would suggest that a normalized dataset corrected for technical effects, such as the dataset of transformed FPKM values, would serve the vast majority of downstream analytical purposes.

Fig R3.1: Test statistic and \log_{10} p-values from t-test using transformed FPKM values and from DESeq2 using raw counts. For FPKM values, its principal component 2 (PC2) was removed, while for raw counts PC2 was adjusted for by DESeq2. Differential expression was tested for anti-CTLA4 resistant versus anti-PD1 resistant samples, as in the manuscript. Correlation of test statistics was 0.79, and correlation of $-\log_{10}$ p-values was 0.57. Dashed black line marks $p=0.05$. Red line passes the origin with a slope of 1.

Fig R3.2: Correlation of t-test and DESeq2 test statistics in uncorrected data, obtained from 50 random categorical variables for differential expression analysis. Red dashed line = median.

Fig R3.3: DESeq2 p-values were added to Fig S4B. Please see legend for Fig S4B. P-values for DESeq2 are often lower than from t-test, resulting in more significantly differentially expressed genes.

Fig R3.4: Gene set enrichment analyses using gene rankings based on t-test (left) and DESeq2 (right) statistics from differential expression analyses between anti-CTLA4 resistant and PD1 resistant cases. Top 10 significant pathways are plotted and are highly similar between methods. Red = up-regulated in anti-CTLA4 resistant samples, blue = up-regulated in anti-PD1 resistant samples.

REVIEWERS' COMMENTS

Reviewer #1 (Remarks to the Author):

There have been some improvements in the Figures. I prefer that Figure legends contain all the details within a figure, but here the authors have chosen to define the abbreviations within the main text. I think this complicates the reading of figures and is not best practice - but I will leave this to the journal to finalise.

Reviewer #3 (Remarks to the Author):

The authors addressed all previous comments and concerns